# Ocean Forecasting for the German Bight: From Regional to Coastal Scales

Emil V. Stanev[1], Johannes Schulz-Stellenfleth[1], Joanna Staneva[1], Sebastian Grayek[1], Sebastian Grashorn[1], Arno Behrens[1], Wolfgang Koch[1], Johannes Pein[1]

[1]Institute of Coastal Research, Helmholtz-Zentrum Geesthacht, Geesthacht, 21502, Germany

*Correspondence to*: Emil V. Stanev (emil.stanev@hzg.de)

**Abstract.** This paper describes recent developments based on advances in coastal ocean forecasting in the fields of
numerical modelling, data assimilation and observational array design, exemplified by the Coastal Observing System for the North and Arctic Seas (COSYNA). The region of interest is the North and Baltic Seas, and most of the coastal examples are for the German Bight. Several pre-operational applications are presented to demonstrate the outcome of using the best available science in coastal ocean predictions. The applications address the nonlinear behavior of the coastal ocean, which for the studied region, is manifested by the tidal distortion and generation of shallow-water tides. Led by the motivation to
maximize the benefits of the observations, this study focuses on the integration of observations and modelling using advanced statistical methods. Coastal and regional ocean forecasting systems do not operate in isolation but are linked, either weakly by using forcing data or interactively using two-way nesting or unstructured-grid models. Therefore, the problems of downscaling and upscaling are addressed, along with a discussion of the potential influence of the information from coastal observatories or coastal forecasting systems on the regional models. One example of coupling coarse-resolution regional
models with a fine-resolution model interface in the area of straits connecting the North and Baltic Seas using a two-way nesting method is presented. Illustrations from the assimilation of remote sensing, *in situ* and HF radar data, the prediction of wind waves and storm surges, and possible applications to search and rescue operations are also presented. Concepts for seamless approaches to link coastal and regional forecasting systems are exemplified by the application of an unstructured-grid model for the Ems Estuary.

## 1 Introduction

Scientific developments are at the heart of newly emerging coastal ocean services supporting blue and green growth (She et al., 2016). Although the volume of coastal ocean observations around Europe is large compared to open-ocean observations,

they alone are not sufficient to fully support the present-day need for high-quality ocean forecasting and monitoring. Therefore, recent practices in this field are based on both observations and numerical modelling. Numerous integrated coastal observing and modelling systems provide not only services but also research advancement; several of them are described by Kourafalou et al. (2015a). The advances in coastal ocean forecasting were recently reviewed by Kourafalou et

al. (2015b). A review on ocean monitoring and forecasting activities in both the open and coastal oceans was presented by Siddorn et al. (2016). The new challenges and trends in this field were recently reviewed by She et al. (2016).

In the present study, we present research issues that are relevant to the pre-operational oceanography in the North Sea and Baltic Sea with a focus on the German Bight, addressing mostly the short-term predictions. The link to the operational forecasting, which is usually performed by authorized state agencies (e.g., the German Federal Maritime and Hydrographic

Agency known as the Bundesamt für Seeschifffahrt und Hydrographie, BSH, and the UK Met Office), is limited in the present study to using freely available data products from their numerical models (Dick et al., 2001; Dick and Kleine, 2007; O'dea et al., 2012) for analyses and inter-comparisons. Skill estimates have been considered in earlier publications (e.g., Barth et al., 2011; Grayek et al., 2011; Port et al., 2011; Stanev et al., 2011, 2015 a, b; Pein et al., 2014; Zhang et al., 2016a), where more details are given about the systems' performance.

The North and Baltic Seas (Fig. 1), which are among the best-studied coastal areas of the world oceans, are also locations of intensive marine use and various activities. This could create possible conflicts while simultaneously attracting wide social, economic and political attention. One extreme event, which attracted societal concern, was the Great Flood of 1962, during which 315 people in Hamburg and an additional 35 in the rest of northern Germany died. Similar devastating extreme events have not occurred since then because of the preventive measures taken, including the implementation of modern prediction

and warning systems.

The need to solve a number of practical problems motivates a serious consideration of the usefulness of coastal research (Storch et al., 2015). One basic issue in estimating the efficiency of present-day coastal forecasting systems is how well these systems benefit from available data (Kourafalou et al, 2015a, b). Therefore, the main goal of this paper is to showcase methodologies that integrate observations and models in coastal areas. The analysis of the synergy of coastal and larger-scale

forecasting systems is our next goal.

Coastal ocean integrated forecasting systems enable monitoring and prediction of the coastal ocean state by accounting for the dominant coastal processes under a wide range of characteristic scales, ranging from the sub-mesoscale to the regional-basin scale. Of major importance are the exchanges along and across the shelf break, storm surges, tides, internal waves, surface waves, fronts, slope currents, estuarine processes, river plumes and suspended sediment dynamics. Near the coast,

biogeochemical processes show large diversity and strong gradients. Possible environmental consequences associated with the transport and accumulation of matter and pollutants require predictions characterized by high confidence and timeliness. The challenge is to use multidisciplinary and multiscale observations, as well as a seamless modelling suite with interacting modules that accurately represent the individual sub-systems: atmosphere, waves, circulation, and biogeochemistry. The inclusion of atmospheric and wave models is important because the air-sea interaction in the shallow coastal ocean can be

traced to the bottom. This specific case differs largely from the known case in the open ocean, where the coupling between the bottom layer and the atmosphere is less direct. Most of the above aspects are relevant to the major topics addressed in the present research and motivate the presentation of the state-of-the-art and recent advancements.

Data assimilation in the coastal ocean is not currently a routine operational practice; however, its use could allow extraction of the most important information from relatively sparse and noisy observations, and this information could be included in numerical forecast models in an optimal way. Therefore, the usefulness of data assimilation in the coastal ocean must be explored. Observation errors are due to instrumental noise, sampling, and possible misinterpretation of measurements. Numerical ocean models are not error-free; errors originate from the incomplete (non-perfect) model physics, insufficient grid resolution, problems with open boundary conditions, and atmospheric or hydrological forcing. Even "perfect" ocean models deviate from reality, which produces a loss of predictability beyond the predictability limit. This limit depends on the geophysical processes. For the synoptic processes in the open ocean, this limit is of the order of weeks to months; for the coastal ocean, it is of the order of hours to days. The loss of predictability (short memory of coastal systems) is associated with nonlinear transfer and the growth of errors. The consideration of these nonlinear effects, which is one of the major subjects in the present study, is addressed using astronomical and shallow-water tide examples.

There are some major differences between the present research and recent studies (e.g., Kourafalou et al., 2015a, b); one of them is that we focus on *only one specific region* (the North Sea and Baltic Sea). There are a number of important dynamic transitions in these specific basins when moving from the regional to coastal scale. The area of major interest in the present study, the German Bight, which is in the south-eastern corner of the North Sea, is where the tidal wave undergoes a pronounced distortion, exemplifying the transition between the regional and coastal ocean. Another interesting (and very close to the German Bight) area of transition between regional and coastal oceans is the multiple strait system connecting the North Sea and the Baltic Sea, which is of utmost importance for the transport of water and salt between the two basins. By focusing on the shallow-tide generation, inter-basin connections and upscaling, we will show how the present study goes beyond other review papers.

The data issues are addressed in more detail in other papers (Baschek et al., 2016). Therefore, we restrict our analysis mostly to numerical modelling, data assimilation, coupled models and seamless modelling in the coastal area, all of which are of fundamental importance to addressing the transition processes between regional and coastal oceans. The presentation of the results is structured into two main sections. Section 2 presents the numerical models used and the specific analyses focused on shallow-water tides and inter-basin exchange. The benefit of using ocean-wave coupled models when predicting extreme events, as well as the estuarine modelling based on unstructured-grid models as a step towards linking estuaries and ocean, is also presented. Section 3 describes new developments in data assimilation in the German Bight, such as assimilation of high frequency (HF) radar data and temperature data from different observing platforms. It also addresses the upscaling problem, which is at the heart of interfacing coastal and regional forecasting systems. Short conclusions and outlook are presented at the end.

## 2 Numerical models: Application to regional and coastal seas

This section presents some physical problems of coastal ocean forecasting and the use of numerical models to solve them. The analysis is concentrated on two major problems: nonlinear (shallow) tide generation and the inter-basin exchange. This selection has been made because, in our opinion, there is a lack of available research with respect to these processes. Their adequate simulation is highly relevant for the data assimilation and upscaling presented in section 3. We first describe some specificities of coastal modelling, followed by a presentation of four forecasting ocean models, which are used in the North Sea. Although these models have a good track record simulating tidally and atmospherically driven circulation, their capabilities to resolve tidal distortion have not been sufficiently addressed. This is the major issue discussed in section 2.2. In section 2.3, the inter-basin exchange is presented using the example of a novel coupling technique for the North Sea and Baltic Sea. Section 2.4 demonstrates the improvement of the forecasting skill for extreme events when using coupled circulation-wind-wave models. Section 2.5, which addresses estuarine seamless modelling and quantifies the pattern of water mass transformation between rivers and open ocean, can be considered as a step towards linking estuarine and regional ocean modelling.

## 2.1 Regional and coastal ocean modelling and forecasting

The resolution of fine spatial scales and high frequencies imposes difficult requirements in ocean modelling. While most of the volume of the open ocean is characterized by a low level of turbulence, the coastal ocean is an essentially dissipative system. Therefore, a more complete representation of the turbulence production and dissipation is needed, along with a deeper knowledge of their temporal and spatial dynamics. The small-scale processes, which are dominant in the coastal ocean, require a deeper consideration of the mesoscale to sub-mesoscale dynamics and their interplay with larger-scale processes. Of particular importance is the improvement in the description of exchanges between the coastal and open ocean, as well as its coupling with estuaries (see section 2.5) and catchment areas. Here, adequate modelling of the fresh water flux is of high priority for the North Sea and Baltic Seas because they are strongly impacted by rivers.

Regional and coastal numerical models require a resolution capacity that is compliant with the dominant spatial scales. It is not only the Rossby radius of deformation, which is between 3 and 10 km in the open Baltic Sea (Fennel et al., 1991) and less than 2 km in the North Sea (Badin et al., 2009), that needs to be resolved but also the estuarine scales, which are much smaller. The areas of drying and flooding present a challenge for the vertical resolution, particularly when geopotential coordinates are used.

It is not only the spatial resolution that maters when moving from the regional to coastal scale but also the details of bathymetry, such as the coastline and bottom roughness, the latter of which can also change in time. Addressing specific processes and their role in the coastal ocean is essential to understand whether we could solve the major problems with the transition between the regional and coastal scales by only changing the resolution. Specific processes, e.g., shallow-water

tides, which are sometimes neglected in global and regional forecasting, dominate coastal ocean dynamics. An additional example that demonstrates the role of surface waves in the coastal zone is presented in section 2.4.

The theoretical developments need to be consistent with the technological advancements in the field of sensors and observational platforms (e.g., high-resolution, wide-swath altimetry and geostationary sensors). One challenge is linking the coastal forecasting to the surface currents that are directly estimated from coastal radar and satellites (see section 3. 2) and high-resolution sea-surface temperature (SST, see section 3.3). In response to these challenges, specific aspects of modelling need to be improved/enhanced. Among the most important developments are the coupling of coastal models with atmospheric and surface wave models (see section 2.4), as well as seamless modelling (see section 2.5). In addition, more flexible coupling is needed between the regional and coastal models, including estuarine models, as well as an adequate presentation of river runoff and its interaction with the coastal ocean (see section 2.5). Advanced numerical schemes and parameterizations are required, in particular for seamless implementation, to simultaneously and adequately resolve multiscale interactions (Zhang et al., 2016b).

For the improvement of model performance, the multi-model approach (Golbeck et al., 2015) has become important; however, the model inter-comparison requires greater focus on physical representativeness. The comparison with data has to be deepened using coastally tailored methods and metrics. Although many developments have been produced in the North Sea high-frequency processes (e.g., tides), tidally relevant metrics are not sufficiently used in the model inter-comparisons.

## 2.2 Numerical modelling of tides in the North Sea: Inter-comparison study

### 2.1.1 Astronomic and shallow-water tides

The tidal dynamics in the deep ocean are almost linear, and the tide can be adequately described using a number of constituents. As demonstrated by Shum et al. (1997) global ocean tide models in the deep ocean agree within 2-3 cm. In shallow water, the tidal dynamics are nonlinear and compound and overtides appear. The nonlinearities are due to the quadratic bottom friction and advection term (Le Provost, 1991). The latter generates overtides (M4) of twice the frequency of the astronomic M2 tide. The friction term is responsible for the generation of the odd harmonics (e.g., M6). These shallow-water tides are very important for the tidal dynamics in some coastal areas. Southwest of the British Isles and in the Irish Sea, their amplitude is comparable with the amplitude of the M2 tide (Andersen, 1999). As demonstrated by Stanev et al. (2015c), the M4 tides in the German Bight cause strong tidal asymmetry. The need for increasing accuracy of tidal predictions in the shelf regions makes it mandatory to account for the higher harmonics and to evaluate the capabilities of ocean models to fully resolve tides (see section. 2.2.4). This is important to adequately simulate dominant coastal processes, such as sediment transport, which is strongly dependent on tidal asymmetry.

There are substantial differences between the quality of model and altimeter data in the open and coastal ocean. Satellite altimetry in the open ocean has high accuracy; the altimeter data close to the coasts are less accurate. The characteristics of models are the opposite: on the continental shelves, coastal models perform better than global ones. In these coastal areas,

the tidal range is usually larger than in the open ocean, and the propagation of tidal waves is more complex because of numerous factors (including the strongly variable bottom friction and the vertical stratification in the regions of fresh water influence, ROFIs). Furthermore, the higher tidal harmonics are difficult to measure with satellites (Andersen, 1999). Therefore, one necessary step is to investigate the adequacy of altimeter data and numerical simulations in the coastal ocean.

Further steps include using data assimilation methods to supplement hydrodynamic models (Andersen et al., 2006; Egbert et al., 2010; Schulz-Stellenfleth and Stanev, 2016).

### 2.2.2 The rationale of inter-comparison: Numerical models used

Although some steps have been performed in the North Sea model inter-comparisons (e.g., one model against another model, see Stips et al., 2014; Su et al., 2014), there is still a demand to: (1) compare more than two models operating in the same

area (Golbeck et al., 2015); (2) consider metrics appropriate for the North Sea; and (3) address model-to-model inter-comparison along with model versus data comparison. The regional models and their setups, which are used in the present inter-comparison study, are briefly presented in the Annexes. Their most important characteristics are summarized in Table 1. The models are the following: Nucleus for European Modelling of the Ocean (NEMO, one setup, FOAM-AMM7, is operational, and the other is run by the authors, NEMO-HZG), General Estuarine Transport Model (GETM), the operational

model of BSH (BSHcmod, one setup run by the BSH is operational, and another (CMOD4) is run by the authors), and the Semi-implicit Cross-scale Hydroscience Integrated System Model (SCHISM), which is an unstructured-grid model. All these models use the primitive equations and have comparable horizontal resolution that ranges from 7 km (FOAM-AMM7) to 2 nm (NEMO-HZG). The horizontal refinement along the coast of the German Bight and in the Danish Straits in SCHISM is approximately 200 m. Most models use terrain-following coordinates (Table 1) of approximately 30 layers; SCHISM uses

up to 59 layers (average number of layers is 29). Tidal forcing is provided either as tidal constituents or a finite element solution (FES2014) (Lyard et al., 2006) or, in the case of SCHISM, as boundary values from FOAM-AMM7. Meteorological forcing is provided from different weather prediction models (Table 1). In BSHcmod, two fine-resolution (~1) models of the same type are two-way nested in the areas of the German Bight and Danish Straits. Except for the BSH operational model, which uses E-HYPE data (Lindström et al., 2010), all models use climatological river runoff data.

The models listed in Table 1 (see also Annexes) are central or related to all applications discussed in the present work (not only for the inter-comparison study addressed in the present section); therefore, a critical examination of their capabilities to reproduce the dominant characteristics of dynamics is needed. Because (1) the tidal forcing is the most important in the North Sea with respect to the amount of mechanical energy provided, (2) the response to atmospheric forcing has been widely addressed for the North Sea (Backhaus, 1989; Skogen et al., 2011; Dangendorf et al., 2014), and (3) the shallow-

water tides, which are very important in the coastal ocean, are not sufficiently addressed in the literature, we focus on the capabilities of different models to adequately simulate tidal distortion.

The models presented above differ not only because the numerics are different. They have different vertical and horizontal resolution, different topography, different boundaries, and different forcing (see the Annexes). The coastline is also

differently resolved in the individual models. Furthermore, one part of the numerical simulations presented below is performed in the frame of the present study. The results from two additional models (FOAM-AMM7 and BSH operational model) are presented using freely available data from the operational agencies. Because the model setups and forcing data differ, it is not possible to trace the strengths and weaknesses of the different models to underlying deficits in numerics, forcing, or representation of physical processes. This is not our aim; we address the question of how several, relatively similar, numerical simulations (some of them operational) compare with respect to tides.

Comparison between numerical simulations and observations has been performed by Andersen (1999), who compared the overtides estimated from altimeter data and those simulated by the model of Flather (1976; 1981). In contrast to that work, we concentrate on the representation of the M4 tide by several models (see Annexes). This provides a good illustration of possible problems associated with nonlinear processes, which many models do not accurately simulate.

### 2.2.3 M2 tides

With the pioneering work of Proudman and Doodson (1924) and many other authors in the 20[th] century, the knowledge of tidal dynamics in the North Sea reached mature status compared to other ocean areas. This is also because the north-western European shelf is the most intensively surveyed ocean area in the world, as far as tidal data are concerned. According to Andersen (1999), a total of 270 coastal and pelagic tide gauges were available on the shelf.

The M2 tide, which is essentially a Kelvin wave, is reproduced in a very similar way by all the models (Fig. 2a). All tidal analyses presented in this figure compare well with the numerous previous numerical simulations cited above, as well as with satellite observations (Woodworth and Thomas, 1990). The highest M2 amplitudes appear in the English Channel. Large magnitudes are also located in the German Bight, where the values simulated in GETM are lower than in the other models.

There are four M2 amphidromic points in the studied model area. The one in the English Channel is close to the British coast, as indicated in the analysis of Chabert d'Hières and Le Provost (1970). The amphidromic point between the British Isles and the coast of the Netherlands is at the same location in all models. Small differences between the NEMO simulations and the simulations of the remaining models exist in the area around the amphidromy in the German Bight. This is attributed to the fact that the minimum depth specified in the two NEMO models along the coast is deeper than in the other models.

The closed boundary in the Danish Straits in FOAM-AMM7 (this model is run for the north-west shelf only) results in the generation of an amphidromic point in the Kattegat. In all other models, the Baltic Sea is included, and the amplitude of the M2 tide continuously decreases while approaching the Danish Straits. This provides the first indication of the importance of the Baltic Sea. One advanced solution to the problem of coupling the two seas is given in section 2.3.

The amphidromic point in front of the Norwegian coast is not identical in all simulations. The M2 phase pattern in BSHcmod and GETM is shifted to the east compared to the rest of the simulations, such that an amphidromic point is not observed in the ocean. The amphidromic point is in the ocean in the two NEMO setups and is almost on the coast in

SCHISM. The most appropriate explanation of these differences is the bathymetry and the resolution of coastline (the most adequate is the resolution in SCHISM).

### 2.2.4 Shallow-water tides

The North Sea shallow-water tides (e.g., M4 and M6) have small amplitudes compared to the M2 tide in the shelf regions
(Fig. 2b). The fact that they are not well resolved by the numerical models and observations could explain the limited knowledge about the spatial patterns of these harmonics. However, most coastal processes are crucially dependent on tidal asymmetry, which is determined by the relationship between the amplitude and the phase characteristics of the overtides and astronomical tides. The comparison of the amplitudes of the individual constituents (not shown here) indicates that the M4 tide is the most important of the nonlinear tides. One area where the amplitudes of the overtides are not small is the English
Channel (Chabert d'Hières and LeProvost, 1970; Andersen et al., 2006), which is supported by the results of all the models (Fig. 2b).

Compared to M2 tides, M4 tides have smaller scales (see Stanev et al., 2015c for the explanation). The M4 tidal amplitudes show very complex patterns, and there are some pronounced differences in all the models (Fig. 2b). This is an indirect demonstration of the differences in the nonlinearities simulated by the individual models. However, a number of features are
qualitatively similar in all models, including the large-scale minimum simulated in the northern area (open ocean). Two M4 amplitude minima are simulated in the English Channel in all models, with slight differences in their position and extension. The low-magnitude M4 area in front of the Elbe and Weser estuaries (identified by the analyses of Stanev et al., 2015c) is simulated by all models; however, its position is slightly different in each model. Good agreement also exists between the simulated amplification of the M4 amplitude in the embayments along the British coast.
One obvious difference between the individual simulations is the representation of the "wavy" patterns along the southern coast. The "protrusion" of the M4 maximum, originating from where the orientation of the southern coast changes to almost zonal, is not well observed in the data provided by the FOAM-AMM7. Comparison with the simulations of NEMO-HZG suggests the horizontal resolution as a cause of this difference.

The overall agreement between the simulations in Fig. 2b and the satellite observations (see Plates 1, 2 in Andersen (1999) is
reasonable. The models addressed in this paper are superior with respect to the relatively old hydrodynamic shelf model of Flather (1976; 1981), which was used by these authors. In contrast to the good agreement between the M2 phases simulated by all models, the phases of the shallow-water tides differ substantially. Because these phases are very important for tidal asymmetry, further attention is needed with respect to the performance of regional models in the coastal ocean. The results of Stanev et al. (2015c), who focused on the role of horizontal resolution for the dynamics of coastal ocean, provide a
plausible explanation of the above problem. With the exception of SCHISM, the models using coarse resolution of several kilometers simulate the tidal dynamics very differently (and perhaps not adequately), particularly the distortion of the tidal signal in the shallow coastal zone of the North Sea.

In conclusion, while the astronomic tides are adequately simulated in the individual models, the simulation of overtides requires further attention, in particular when developing coastal applications. This is one of the reasons the transition from the regional to coastal scales presented in the following sections is focused on the German Bight. Further model inter-comparisons should quantify the numerical simulations against available data from remote sensing, tidal gauges and bathymetry. The use of time-referenced bathymetry is also important because of the rapid migration of the bottom channels in the Wadden Sea (Jacob et al., 2016). These authors demonstrated that the model response to bottom changes is strong for the M4 tide, providing further motivation to deepen the understanding of the dynamics of shallow-water tides.

## 2.3 The inter-basin exchange: Two-way nested NEMO

### 2.3.1 Rationale for the study

The Danish Straits are fundamentally important to the exchange between the North Sea and the Baltic Sea (Sayin and Krauß, 1996), providing a major control for Baltic Sea stratification (see, e.g., Meier and Kauker, 2003; Döös et al., 2004; Feistel et al., 2006, and the references therein). Because of the very narrow cross-sections and complex topography with small-scale features, the dynamics are dominated by small-scale motion. Therefore, this geographic area is an excellent location to illustrate the role of small-scale processes in these *"choke points"* in the dynamics of regional seas, which fits the major topic of the present study: the transition from regional to coastal scales.

The most commonly used approaches to address the complex Danish Straits' bathymetry in nested numerical simulations were addressed by She et al. (2007), along with the influence of bathymetry on the salt- and freshwater flow rates. However, this and other previous studies did not address the nesting procedures and technicalities that have to be applied to the transition area between the North Sea and the Baltic Sea to adequately resolve the water and salinity exchange. One solution would be the use of a standard two-way grid refinement tool for the NEMO framework (e.g., AGRIF, Adaptive Grid Refinement in FORTRAN). Current practices demonstrated that this was an effective tool for horizontal grid refinement (see, e.g., Laurent et al., 2005; Cailleau et al., 2008; Jouanno et al., 2008). However, to our knowledge, no applications for vertical grid refinement exist. The novel development here is the two-way nesting method, which enables the use of different vertical discretization in the individual nests of the North Sea–Baltic Sea NEMO. The use of different vertical grid types (σ-levels and z-levels) in the different parts of the nested models is proposed as a step forward from the system described by She et al. (2007).

NEMO-HZG, presented in section 2.2.2 (see also Annexes), is used in several configurations. They include three nested areas: one for the North Sea (red and green areas in Fig. 3a), one for the Baltic Sea (green and blue areas) and one for the Danish Straits (green area in Fig. 3a). Thus, the North Sea and Baltic Sea models overlap over the transition area between the two seas. The horizontal resolution of the North Sea and Baltic Sea models is 2 nm. In the vertical direction, the North Sea model uses 21 σ-levels and the Baltic Sea model uses 35 z-levels. The choice to use z-coordinates in the Baltic Sea was

made to avoid possible problems caused by pressure gradient errors in terrain-following coordinate systems when density stratification is very strong.

### 2.3.2 Description of model coupling

The two coarse-resolution models exchange data at their outer open boundaries. At the Baltic Sea boundary, the North Sea
model uses boundary forcing provided by the Baltic Sea model, which is interpolated onto the North Sea grid. In the same way, the Baltic Sea model receives boundary forcing data at its western boundary from the North Sea model. The first two panels in Fig. 3c show the simulated salinity and currents along the transect shown in Fig. 3b. As expected, the representation of the dynamics in the transition zone of the two models is not identical; the z-level model (in the Baltic Sea) reproduces the estuarine circulation in the straits, while the σ-level model (in the North Sea) reveals much weaker
stratification. The physical parameterizations in the two models are the same.

To quantify the effect of the horizontal resolution in the straits on the model performance, another experiment was performed, in which the interplay between the North Sea and Baltic Sea models was different: (1) data are exchanged not only via the open boundaries (as in the case considered above), and (2) a Danish Straits model with a finer spatial resolution of 0.5 nm in the horizontal and 35 terrain-following σ-levels in the vertical is included. The proposed method distinguishes
between 'parent' and 'child' nests or models. The child nest (the Danish Straits model) receives its boundary forcing from the parent nests (the North Sea and the Baltic Sea models) at its open boundaries. In the following "assimilation" step, the child model exports its data onto the overlapping area of the two parent models using a data assimilation approach. This second step is the main difference between the proposed nesting and the classic method. The difference is the handling of the information flow from the child to the parent nests, which enables gradual upscaling (see also Schulz-Stellenfleth and
Stanev, 2016) while maintaining the overall dynamic consistency of the parent nest. Another basic difference in the existing practices is that the "child nest" has "two parents", in our case, the North Sea (red and green) and Baltic Sea (green and blue) models, which do not directly communicate.

The different models are synchronized in the following way: The coarse-resolution model runs are segmented in a one-day hindcast and a one-day forecast phase. During the hindcast phase, the coarse-resolution models receive enhanced fine-
resolution information over the whole domain of the fine grid. During the forecast phase of the coarse-resolution model, no nudging is applied, and the models run in a free prognostic mode. In the next cycle of the coupling procedure, the fine-resolution model is re-run for the whole hindcast-forecast period of the coarse-resolution model using the interpolated coarse-resolution output as the boundary condition.

### 2.3.3 Analysis of the simulations

The two bottom panels of Fig. 3c depict the simulated salinity and velocity fields from the two Baltic Sea parents and child runs during a small inflow event on 16.09. 2010. The first illustrates how the Baltic "parent" sees the transition zone; the second illustrates how the "child" nest sees the transition zone. Because of this specific nesting, the Baltic Sea "parent"

receives some features from the North Sea counterpart via the "child" model (see the third plot in Fig. 3c). The fine-resolution nest shows some displacement of the estuarine front and increased bottom salinity in the vicinity of the Darss Sill (at approximately 80% of the transect length) and thus increased stratification.

The differences between individual simulations can be explained by the changes in the secondary circulation (Fig. 3d), resulting in a stronger vertical current component, which brings saltier water originating from the Kattegat (between approximately 10% and 45% of the transect length). These subtle processes are challenging to simulate on a coarser-resolution grid. Although the overall salinity and current patterns along the cross section are comparable in the two simulations (Fig. 3c), the salinity transport along the track deviates significantly in the individual models. Comparison of the "Baltic Sea parent (z-level; coarse)" with the "Baltic Sea parent (z-level; coarse) + assimilation" (the third and fourth plots in Fig. 3c) shows that the proposed procedure helps to reduce this deficiency without a negative impact on the consistency in the coarse-resolution simulation.

When nesting models with different resolutions, one could expect a distortion of model performance at the nested boundaries, which might propagate in the both directions. To check this issue, we first compare the tidal signal in two simulations focusing on the M2 tide (Fig. 4a, b). The horizontal patterns of the amplitudes in the reference model and in the fine-resolution nests are similar (note that the land-sea mask is not the same in the two models). The results resemble those presented in Stanev et al. (2015a), illustrating that the tidal amplitudes (Kelvin waves) are higher along the right coast. The relatively higher amplitudes in the reference run are explained by the difference in the topographies of the two models. With increasing realism of the topography in the fine-resolution nest, the amplitude of the M2 tide decreases, tending to the values presented by Stanev et al. (2015a), who used 1 km horizontal resolution in the transition area. Because the penetration of the tidal wave into the Baltic Sea is damped when using fine-resolution topography, the relatively large amplitudes simulated in the reference run in the area of Lübeck Bay are reduced when the transition area is simulated with finer resolution. The phase lines in the two simulations show qualitatively similar configurations; the differences between the two simulations do not exceed 10-15 min. These almost negligible phase differences indicate that the proposed method does not distort processes with fast time scales, such as tidal wave propagation.

The sensitivity to salinity in the transition area between the North Sea and the Baltic Sea is illustrated in Fig. 4c, d for two depths. This plot shows first that although the temporal variability is similarly represented in the reference simulation and in the simulation using z-coordinates in the Baltic Sea, the differences are not negligible. This is an instructive result when determining the appropriate vertical resolution for models that aim to simultaneously resolve the two basins. The role of the horizontal resolution is even more important. Although the courses of the salinity curves follow similar variability, which is a response to the atmospheric forcing (for the analyzed period, the periodicity is approximately weekly), the amplitudes differ largely. During the strongly mixed conditions shown in Fig. 4c, d, with almost equal salinities at 7 and 17 m shortly before 11.09.2010, the differences between simulations in all models are minimal. These results prove that the model performances in the transition zone could differ depending on the different inflow-outflow conditions.

The nested area considered above acts as the mouth of the Baltic Sea, which is one of the largest estuaries worldwide. We demonstrated that simulations in this area with coarse resolution misrepresent important characteristics of the salinity front (position and stratification), as well as the secondary circulation, which could affect the characteristics of the Baltic Sea conveyor belt (Döös et al., 2004) and the outcome of long-term simulations. A different approach to address the transition area linking the Baltic Sea and the North Sea was proposed by Zhang et al. (2016a) using unstructured grids.

## 2.4 Coupled circulation and wave models: A step towards improving the forecasting skill for extreme events

### 2.4.1 Model description

In the last decade, the Northern European coasts have been affected by severe storms that have caused serious damage to the North Sea coastal zones. Additionally, human activities, e.g., the offshore wind power industry, oil industry and coastal recreation, require information about the sea state in the coastal ocean with high resolution in space and time. There is a consensus that high-quality predictions of extreme events, such as storm surges and flooding caused by storms, could contribute substantially to preventing or minimizing human and material damages and losses. Therefore, reliable wave forecasts and long-term statistics of extreme wave conditions are of utmost importance for coastal areas. In many coastal areas, the need for reliable risk assessment increases the demand for precise coastal predictions. This section demonstrates that precise predictions cannot be achieved without considering the wind-wave–current interaction.

Oceanic flows can be strongly forced or modified by waves, particularly in the nearshore and coastal ocean (Lentz et al., 2008; Longuet-Higgins, 1970; Newberger and Allen, 2007). The interactions between surface waves and ocean currents control the boundary fluxes, momentum and energy exchange between the atmosphere and the ocean, as well as important processes within the water column. Rascle and Ardhuin (2009) demonstrated that a proper representation of the near-surface currents and drift requires the introduction of wave effects, in particular, Stokes drift and wave-induced mixing. The coupling of wind waves and circulation models is intimately related to the fundamental issue of the air-sea interaction and, in more specific terms, the improvement of atmospheric forcing for ocean models. Furthermore, the German Bight is dominated by strong tidal currents that exceed 1 m/s in some areas; therefore, the nonlinear feedback between currents and waves plays an important role in this area. As demonstrated in this section, wind-wave–circulation coupling must be accounted for during extreme events, which enhance the nonlinear interactions.

The analyses presented below are based on the wave model WAM (the WAMDI group, 1988; Guenther et al. 1992). This model is used in COSYNA to forecast the sea state in the southern North Sea (the model domain is shown in the upper-left zoom of Fig. 1). WAM is a 3[rd]-generation surface wave prediction model based on the action density balance equation in frequency/direction coordinates. Multiple nesting is possible. The model is forced by time series of the surface (10 m) wind, wave spectra at the open boundaries, currents and water level. The output of this model includes the significant wave height, wave periods (Peak, Mean, Tm1, Tm2), wave direction, directional spread and wave spectra (frequency-direction). In WAM,

the thickness of the water column and/or current fields can be non-stationary, grid points can fall dry and refraction due to spatially varying currents is represented.

As a counterpart of WAM, we use the German Bight model described by Staneva et al. (2009) and Stanev et al. (2011). This model is based on GETM (see Annex 5.2) and has horizontal resolution of 1 km (see also Fig. 6 of Stanev et al., 2011). The
model coupling can be achieved at different levels of complexity. Staneva et al. (2016) used an off-line coupled system for the German Bight that considered: (1) the effect of currents on waves and (2) the effects of waves on the upper ocean dynamics, in particular, on the mixing and drift currents. In the present study, the wave model includes a revised approach for wave breaking in coastal areas and modified wave growth in the source term for the wind input. The GETM-WAM coupling is fully two-way and uses the coupler OASIS3-MCT: Ocean, Atmosphere, Sea, Ice, and Soil model at the European
Centre for Research and Advanced Training in Scientific Computation Software (Valcke et al., 2013). The original version of GETM was modified to account for the depth-dependent radiation stress and Stokes drift. The terms were calculated from the integrated wave parameters according to Mellor (2011) and Kumar et al. (2011). The gradients of the radiation stresses were implemented as additional explicit wave forcing in the momentum equations for the horizontal velocity components. The Stokes drift components are subtracted from the wave processes to transfer the problem to the Eulerian framework.
Moghimi et al. (2013) studied the effects of two approaches (radiation stress, Mellor, 2011; and vortex force, Ardhuin et al., 2008) using GETM–SWAM coupled models and showed that the results for the longshore-directed transport are similar for both formulations. Recently, Aiki and Greatbatch (2013) showed that radiation stress parameterization is applicable for small bottom slopes. Grashorn et al. (2015) quantified the applicability of this formalism for the German coastal zone.

The necessary wave-state information required to compute the divergence of the radiation tensor in the momentum equations
is provided by WAM. WAM also provides information about the dissipation source functions (wave breaking and white capping, as well as bottom dissipation) to the turbulence module GOTM, where it is used to calculate the boundary conditions for the dissipation of turbulent kinetic energy and the vorticity due to wave breaking and bottom friction (Pleskachevsky et al., 2011). Additionally, bottom friction, depending on the bottom roughness and wave properties, has been implemented (Styles and Glenn, 2000). Table 2 gives a summary of the improved model performance with respect to
prediction of the sea level, which is the main variable considered below in the analysis of extreme surges in the German Bight. The quantification of performance demonstrates that in a large number of coastal locations, both the root mean square (RMS) difference and bias between the model estimates and observations are significantly reduced because of the improved representation of physics. This provides an answer to the question addressed in section 2.1 of whether one could adequately make the transition from the regional to coastal scale by only increasing the horizontal resolution. Obviously, the dominant
processes in the coastal ocean also need to be accounted for.

### 2.4.2 Validation and sensitivity study

Four experiments with the standalone wave model are conducted to determine which physical processes are important in a very shallow area near the coast. The first experiment is the default setup of the wave model without wave breaking

(no_wb). The second experiment considers wave breaking (wb+). The new extended formulation of wave breaking avoids simulating unrealistically high waves. Since the waves are very sensitive to the driving wind fields, two further experiments are performed by changing the wave growth parametrization in the wind source term. The Miles parameter in the wave growth parametrization includes a constant $\beta_{max} = 1.2$, which is well adapted to the wind fields generated by the operational integrated forecasting system (IFS) of the ECMWF. The driving forces of the wave model are the COSMO-EU wind fields of the DWD, in which the wind speeds are usually higher than those of the IFS. Therefore, in two additional experiments, the sensitivity to $\beta_{max}$ is tested to achieve better adaptation to the wind fields of the atmospheric COSMO-EU model. The values are 1.05 (bm1.05) in the first and 0.95 (bm0.95) in the second experiment. Wave breaking is also considered in the two experiments.

In addition to the model validation presented by Staneva et al. (2015), we present a representative example of the validation of the WAM in the German Bight during an extreme event. On the 05.12.2013, the severe storm Xavier hit the coast of Germany, with westerly winds greater than 30 m/s. The significant wave height reached ~8 m at the peak on the December, 6[th] at 03 UTC. An illustration of the model performance is given in Fig. 5 for the Elbe data station (see the location in Fig. 1) for the beginning of December 2013. Evidence of the change in surface currents, as observed by HF radar, is given in section 3.2.

The sensitivity experiments help to identify the contribution of different physical mechanisms. Wave breaking is very important in the shallow coastal area. Neglecting it results in severe overestimation of the observed significant wave height at the peak of the storm (see the black curve in Fig. 5 with unrealistically high values). When the new, improved option for wave breaking is taken into account, the model shows much better agreement with the observations (orange curve). The best match between the modelled and measured data is achieved with the additional use of the modified wave growth parametrization, which ensures a more appropriate adaptation to the driving wind fields (blue curve). This result is supported by the statistics (the RMS error, RMSE) and the bias between the observations and the coupled model) at the locations of three buoys (Table 3), demonstrating the skill of the new modified version of the COSYNA wave model.

The gradient of the radiation stress serves as an additional explicit wave forcing term in the momentum equations for the horizontal velocity components. The transfer of momentum by waves is important for the mean water level setup (Fig. 6a) and for the alongshore currents generated by waves in the surf zone.

### 2.4.3 Response patterns in extreme conditions

The horizontal patterns of the maximum difference between the sea level in the coupled wave-circulation and standalone circulation models reveal the impact of wave-induced forcing on sea level. These differences are computed at each grid point during the period of an extreme event (storm "Xavier"), which occurred from 02.12.2013 to 10.12.2013. The patterns (Fig. 6) show that the simulated surge differences between the coupled and circulation-only models are more pronounced along the coastal areas of the German Bight. The maximum difference is approximately 40 cm in the North Frisian Wadden Sea region (Fig. 6a). In the open sea, the differences in the simulated surge characteristics are negligible. The sea-level variability

for the Helgoland tide gauge (Fig. 6b) indicates that during normal meteorological conditions, the coupled and non-coupled models fit the tide gauge data. However, during the storm Xavier, the sea level predicted by the pure hydrodynamic model is underestimated by more than 30 cm. The sea level predictions of the coupled model are closer to the tide gauge measurements (compare the green symbols and black lines).

The basic conclusion of this section is that the large differences between the numerical simulations in the coupled and un-coupled models indicates that accounting for the wind-wave effects in the three-dimensional hydrodynamic model improves the predictions of the water level in shallow coastal waters. Predictions of storm events with coupled models could be of utmost importance for many coastal applications addressing risk analyses (e.g., offshore wind industry, oil platform operations) where higher accuracy is needed. This justifies the consideration of waves in operational forecasting. Similar

new developments will improve the use of atmospheric forcing for wave and ocean models and in the long run, and will result in the development of operational coupled models for the coastal zone.

The uncertainties in storm surge predictions and the quantification of associated coastal hazards is of great interest for both short-term forecasts and climate change analyses. Although storm surge forecasting technology is gradually improving, the real-time assessment of the storm surge and inundation area fails to satisfy various demands, particularly for real-time storm

forecasting. To reduce the uncertainty of forecasts, knowledge about the processes, such as tide-wave-surge interactions, must be improved. Improved weather forecasting and more adequate coupling between the atmosphere, ocean and waves should further reduce the uncertainty. The use fine horizontal resolution in near-coastal areas, which recently became possible because of the availability of improved computational resources, has proved beneficial. The results of our experiments showed that the wave-dependent approach, which is not routinely used operationally, yields an ~30% larger

surge during the period of "Xavier".

## 2.5 Cross-scale modelling: A step towards linking estuaries and open ocean

Coasts and estuaries present a challenging research case for environmental studies and applications. One region of interest to science and society is the Ems Estuary (see, e.g., Chernetsky et al., 2010), which was shaped to a great degree by storm surges during the Middle Ages and is currently entirely surrounded by dikes, with its river area protected by a storm surge

barrier. Intense economic exploitation, especially regular dredging of the navigation channel, has led to very high concentrations of suspended sediment in the tidal river (de Jonge et al., 2014).

There is also profound interest in the physical oceanography of estuaries, which constitute the border between the deep ocean and land, as well as between salt water and freshwater (Dyer, 1973). Estuaries are subject to vigorous tidal currents and highly variable water levels and feature a large number of physical processes that interact to generate complex dynamics.

Some key questions are how the estuarine dynamics interact with the larger-scale dynamics and what the challenges for numerical modelling are. To address these questions, we consider the Ems Estuary, which has a mean river discharge of ~80 m³/s. The volume of fresh water per tidal period corresponding to this value is approximately 20 times smaller than the tidal prism. Furthermore, the maximum fresh water flux could exceed 300 m³/s; thus, the ratio between the tidal prism and the

volume of fresh water per tidal period varies largely in time. One could expect that the dynamics can change strongly in space and time; however, this is still not known for this estuary. The pathway of fresh water penetration into and beyond the tidal river is addressed in this section.

The numerical modelling of realistic estuaries (with realistic topography, coasts, surface and open boundary forcing, river runoff forcing, and adequately resolved baroclinicity and 3D turbulence) is not widely used in estuarine-coastal-ocean forecasting. One reason is that the available models do not sufficiently address the cross-scale interactions between the estuaries and open ocean. Part of this fundamental problem is the transformation of river water. In many ocean models, river runoff is specified as a point source, which does not accurately reflect the real process of fresh water transformation in the coastal ocean. The structured-grid models addressed in previous sections, and many additional similar models, either do not fully resolve estuarine processes or do not adequately interface with estuarine models. Thus, the formulation of their river boundary conditions is not optimal.

We address the interfacing of the estuarine environment with the larger-scale German Bight models using unstructured-grid models. The focus is on the area where the largest transformations of water masses are observed, which includes the area of the salinity front. The model area (Fig. 7a) extends far beyond the Wadden Sea. In these areas, the coastal circulation models, even with 1 km resolution, usually have problems with their performance (see sections 3.2 and 3.3, as well as Fig. 15 of Stanev et al., 2011).

Unstructured-grid numerical models show good skill with respect to subtidal, tidal and intermittent processes in coastal and estuarine environments (Zhang and Baptista, 2008; Zhang et al., 2016b, see also Annex 5.4). The model of the Ems Estuary for the area shown in Fig. 7a, which was developed by Pein et al. (2014) using SCHISM (see Annex 5.4), can be used as a tool to address a number of research and practical questions regarding the function and the physical peculiarities of this specific region. In the present study, new analyses based on simulations with this model are presented, along with an illustration of its performance for assessing tidal distortion (and the resulting asymmetry of tides). These processes are fundamental not only for sediment dynamics but also for the mixing patterns and propagation of fresh water into the open ocean.

The dominant dynamics in the Ems Estuary are induced by tides (van de Kreeke and Robaczewska, 1993). The M2 amplitude (Fig. 7a) is small near the western open boundary of the model area and grows to almost 1 m at the easternmost barrier island. In the Ems Estuary, it increases with the convergence of the topography, i.e., towards the tidal river and Dollart Bay. Reaching a maximum of approximately 1.5 m at the entrance of the tidal river, the main lunar tide is damped near the head of the estuary.

The M4 tide has smaller amplitude, which increases continuously towards the tidal weir (Fig. 7b). The large amplitudes observed in Dollart Bay are indicative of the enhancement of this periodicity caused by the nonlinear response associated with hypsometry (Stanev et al., 2003). Near the tidal weir, the M4 tide reaches half the amplitude of the M2 tide. The ratio between the M4 and M2 tidal amplitudes is approximately 0.1 in the western part of the open sea and the outer estuary,

indicating strong distortion of the tidal wave. The maximum flood currents are much stronger than the ebb currents in most of the channel, revealing a pronounced flood-ebb asymmetry.

This short description highlights the ability of the Ems Estuary model to capture the key processes of the estuarine dynamics (see, e.g., Geyer and MacCready, 2014) in this specific marine environment. Pein et al. (2014; 2016) did not sufficiently

address the transformation of fresh water from the tidal river to the open ocean. The vertical mean salinity averaged for one spring-neap period (Fig. 7c) reveals a relatively simple distribution: (1) very low values along the tidal river up to Dollart Bay, (2) a rapid increase in salinity between km 50 and km 20, (3) pronounced lateral gradients near km 20, with lower salinity more aligned to the right coast when looking towards the ocean, and (4) much lower gradients around and beyond the barrier islands.

The empirical orthogonal function (EOF) analysis of the model data demonstrates that approximately 60 percent of the variability of surface salinity is described by the first mode (Fig. 7e) and an additional 20 percent is described by the second mode (Fig. 7f). The pattern of EOF-1 closely follows the tidal channel and tidal river, while EOF-2 reveals a pattern that is attached to the right coast, as in many ROFIs. The pattern of EOF-2 has some similarities with the averaged distribution of salinity (lower salinity values are observed along the right coast), but the shape is much more pronounced, as shown in the

comparison with Fig. 7c.

The spectral analysis of the principle components (PCs) reveals, along with the dominant M2 maximum, clear maxima corresponding to the shallow-water tides (M4) and spring-neap period (Fig. 7d). While the basic spectral maximum in PC-1 is semidiurnal, the basic spectral maximum in PC-2 is during the neap-spring period. This 14 day periodicity indicates that the horizontal pattern in Fig. 7f reflects the spatial variability of salinity associated with the spring-neap variability.

The relationship between the distribution of salinity and spring-neap variability has been known for a long time (see Stanev et al. (2007; 2015c) for the modelling aspects of this issue in the German Bight estuaries); however, its dominant role in the area of the Ems Estuary has not been previously studied. The difference between the simulated surface salinity during the neap and spring tides (Fig. 8a) is consistent with the general knowledge that the less energetic tidal oscillations during the neap tide result in more stratified conditions. This figure also shows that the difference pattern is attached to the right coast,

similarly to the EOF-2 pattern (Fig. 7f). This demonstrates that the export pathways of fresh water from the estuary are not identical during the two tidal phases and that the differences can be traced beyond the back-barrier islands. The scales of the spatial patterns dominating the fresh water intrusions cannot be resolved by the larger-scale models, demonstrating the need to either interface the regional predictive models with estuarine models or develop unstructured-grid models for larger-scale predictions. Thus, without resolving these patterns in the regional models, one cannot appropriately prescribe the river

boundary conditions.

To demonstrate how the fresh water propagation into the open sea is dependent on river runoff, two simulations in which river runoff is very different are compared. During11.06.2012-15.06.2012 and 11.02.2013-15.02.2013 the fresh water flux amounted to 40 m$^3$/s and 140 m$^3$/s, respectively. The difference in the two runoff situations resulted in very different patterns (Fig. 8b, d). The largest differences between the two runoff situations, as observed in the salinity field, occurred in Dollart

Bay and in the lower part of the tidal river, which is the position of salinity front, as shown in Fig. 8c. The difference between the two experiments changes as a function of runoff (in the considered case approximately 10 km); the difference is very strong (up to 6 ppt) in the front area (Fig. 8d, see also Fig. 7c, where the km-line measuring the distance along the channel is shown). It is clear that the proper definition of the river boundary condition for larger-scale models requires
deeper consideration (or appropriate parameterization in the coarse-resolution models) of the processes in the lower part of the tidal river.

The issue of the predictability of estuarine dynamics, and in more specific terms, the skill of the predictions, is minimally addressed in the oceanographic literature. Based on the above results, one could ask the questions: (1) How predictable are the dynamics in the Ems Estuary? (2) Can the predictability be enhanced by using observations? As shown in Fig. 7a, b, the
M2 and M4 tides have clear horizontal patterns, which are consistent with the observations, as shown in Pein et al. (2014). One can expect that provided that the open-ocean forcing for the area is known, tidal predictions could be used to infer the temporal-spatial changes in the estuary. This cannot be performed for the entire area using point-wise observations alone. An initial step was proposed by Pein et al. (2016), who investigated how well the estuarine state (e.g., salinity) can be reconstructed using synthetic (provided by the model) observations as a first step towards observational network design. This
is justified by the fact that the amount of continuously available data that could be used for statistical assimilation is very limited.

## 3 Data assimilation

Data assimilation in the coastal ocean is a relatively new research field, and its usefulness is still not fully understood. Coastal ocean models are usually implemented in relatively small areas, and their performance is strongly dependent on
open-ocean and meteorological forcing. The limited amount and sometimes bad quality of the data limit the success of data assimilation and make the outcome of data assimilation questionable. This section is focused on the German Bight, and only one model, GETM, is used. This is the same 1 km circulation model presented in section 2.4, which is one-way nested in the regional GETM (see Annex 5.2 and Staneva et al., 2009; Stanev et al., 2011). Two examples of data assimilation are considered: (1) assimilation of HF radar data in section 3.2 and (2) assimilation of SST from different platforms in section
3.3. Possible inconsistencies between the forcing data and assimilated observations could become critical, particularly when addressing the interplay between regional and coastal prediction systems. Therefore, a theoretical framework to understand the problems arising from non-seamless modelling is presented in section 3.4, which is focused on the upscaling and downscaling problems.

### 3.1 Data assimilation in the coastal ocean

In recent years, ocean data assimilation and forecasting has reached a high level of maturity (Chassignet and Verron, 2006). One example is the Global Ocean Data Assimilation Experiment (GODAE), where several systems were developed and

operated by the Australian Bureau of Meteorology (BLUElink Ocean Data Assimilation System, BODAS), the Jet Propulsion Laboratory (Estimating the Circulation and Climate of the Ocean, ECCO), the UK Met Office (Forecast Ocean Assimilation Model, FOAM), other Copernicus systems based on the NEMO VARiational data assimilation (NEMOVAR), and others (Cummings et al., 2009). These GODAE systems assimilate various measurements, such as sea-level anomaly data provided by satellite altimeters; subsurface temperature and salinity data from Argo floats, moored and drifting buoys, expendable bathythermographs (XBT), and conductivity-temperature-depth (CTD) recorders; *in situ* and satellite sea-surface temperature data; and satellite-derived sea ice concentration and drift data.

Ocean data assimilation techniques have been applied for operational forecasts, error analysis, parameter optimization, ocean process studies, and observational network design. Compared to the methodologies used in meteorology and global oceanography, coastal forecasting techniques are at an early stage of development. This is because the specific problems of ocean data assimilation in coastal ocean are challenging and are not sufficiently addressed in global or regional ocean data assimilation. This motivates us to formulate specific coastal problems and to illustrate solutions for some of them.

The complexity of data assimilation in the coastal ocean is increased by the vast range of phenomena and the multitude of interactive scales in space and time (DeMey et al., 2009; De Mey and Proctor, 2009, Korres et al., 2012). The spatial and temporal resolution required for realistic coastal predictions is much higher than the resolution required for the deep ocean. Processes that are sometimes disregarded in open-ocean data assimilation, such as tides and the high-frequency barotropic response to atmospheric forcing, are dominant in the coastal ocean. The small temporal scale (hours) and horizontal scale (hundreds of meters) are computationally and scientifically challenging for data assimilation.

The diversity of methods used to assimilate data in coastal models reflects the complexity of coastal processes and the status of forecasting systems, which face research challenges. Efforts are underway to test and improve the quality of data assimilation; one example for the region addressed here is presented by Stanev et al. (2011). Several problems associated with coastal data assimilation are listed below.

1. *The variables of interest* for coastal applications include the same physical properties as in the open-ocean models in addition to near-bottom currents, which are important for sediment transport, and a large number of biogeochemical properties. This greatly increases the number of variables and the complexity of the models and the assimilation schemes. Short time scales (e.g., minutes to hours for tides) increase the demand for both high-quality observations and specific data assimilation schemes.

2. *A vigorous adjustment process* arises in sequential data assimilation, when the models are restarted (e.g., Malanotte-Rizzoli et al., 1989). A too frequent assimilation of observations can lead to a situation where the assimilation degrades the model results due to the high-frequency perturbations generated by the assimilation (Talagrand, 1972). One approach to overcome this problem is illustrated in the following sections.

3. *The data and observational platforms* differ from those in the open ocean. For example, satellite altimetry does not fully resolve all important coastal-ocean scales, and data from profiling floats are not available in the shelf seas. However, data from HF radar and acoustic Doppler current profiler (ADCP), sea-level data from coastal tide gauges and bottom pressure

gauges, water properties from fixed data stations and ferries, gliders, and AUVs provide new perspectives. In particular, the assimilation of altimeter data must also account for the aliasing of the tidal signal, which can be compensated by using the synergy between the altimeter, tide gauge and HF radar data.

4. *The complex physics* in the coastal zone complicate the assimilation of data and necessitate resolution of the whole spectrum of free-surface variation (tides, storm surges), multiple scales, friction and mixing effects and associated tidal straining and fronts, dependency of the solution of small-scale bathymetric channels and variations of bathymetry (which is not well known, see Jacob et al., 2016), control of the straits for the inter-basin exchange and of the inlets for the exchange between tidal flats and open ocean, drying and flooding. The situation is further complicated by the complex nonlinear processes (e.g., creation of overtides), the strong coupling of the variability at different frequencies, and the relatively "short memory" of the physical processes.

5. *The observation error specification* is extremely challenging in the coastal zone; specific coastal processes necessitate the use of dynamically consistent error-prediction schemes (Stanev et al., 2015b). Most existing assimilation schemes assume unbiased observations with Gaussian noise, which is often unrealistic. For many coastal observational platforms, the determination of errors is difficult because some platforms, e.g., satellite altimeters, have larger errors in the coastal zone.

6. *The coupling of coastal and deep ocean models* is not a well-solved problem. Most coastal models are one-way nested; the models solution is strongly dependent on the boundary forcing originating from larger-scale models. Two-way nested models enable (assimilated) information from coastal observations, which is usually not assimilated by the larger-scale forecasting systems, to be propagated out of the coastal region. The resulting upscaling capability could be beneficial for regional models.

## 3.2 Assimilation of HF radar data

### 3.2.1 COSYNA surface currents: blending surface currents from HF radar observations and numerical modelling

Hundreds of HF radar systems have been installed worldwide in both operational and experimental modes (Harlan et al. 2010; Willis, 2012). With their large area coverage, high resolution in time and space, and long-term operational capabilities, the radar systems have enhanced the coastal ocean monitoring capabilities for surface currents (Paduan and Rosenfeld, 1996) and have enabled the development of new data products. The value of HF radar data for the investigation of circulation in the German Bight was demonstrated by Carbajal and Pohlmann (2004) and Port et al. (2011). Currently, an observation network of three Wellen Radars (WERAs) in the German Bight operates as part of the COSYNA pre-operational system (see Fig. 9 for their locations). Each radar measures the radial components of the surface currents (Stanev et al., 2015b; Baschek et al., 2016). By combining the data from the three radars, one can compute the surface current vectors over a relatively large region (Fig. 9a). This has been performed in the past for moderate-wind conditions. Because we have addressed the forecasting capabilities for weather conditions dominated by extreme events in section 2.4, we show in this figure the de-tided signal capturing the change in the surface current during 05.12.2013-06.12.2013 caused by the storm "Xavier"

propagating over the North Sea and Baltic Sea from 02.12.2013 to 10.12.2013. For the period presented in Fig. 9a, the surface current underwent a substantial change, resulting in a pronounced setup in the coastal zone (see also section 2.4). It is therefore challenging to explore the usefulness of these data in pre-operational forecasting.

Most previous efforts to assimilate HF radar into numerical models used (1) filtered data and (2) the classical Kalman
analysis method, where observations and numerical simulations are combined at individual time steps. The techniques applied include optimal interpolation (e.g., Breivik and Saetra, 2001), variational approaches (e.g., Sperrevik et al., 2015) and empirical methods without the use of a free model run (e.g., Wahle and Stanev, 2011; Frolov et al., 2012). Extensive validation of the performance of the HF radar data assimilation is provided by Barth et al. (2008) using ADCP data, Yaremchuk et al. (2016) using data from drifters, and Sperrevik et al. (2015) using both drifters and ADCP data.

Classic assimilation filters, where the analysis is performed based on observations and model data at a certain time step, are not optimal for areas, such as the German Bight, that are strongly dominated by tides and where the predictions are needed at intra-tidal time scales. The assimilation of HF radar data is not a trivial task because of irregular data gaps in time and space, inhomogeneous observation errors, and inconsistency between boundary forcing and observations (Breivik and Saetra, 2001). In the following, we briefly present a novel method that blends models and HF radar data in an efficient and
dynamically consistent way.

The method developed by Stanev et al. (2015b) uses a spatiotemporal optimal interpolation (STOI) filter to improve short-term hindcasts and forecasts of the surface currents. In the proposed approach, model simulations from a free run and radar observations acquired over periods of at least one tidal cycle are blended using the Kalman analysis equation. The proposed data assimilation approach is similar to the methods described in Barth et al. (2010) and Sakov et al. (2010) but uses a
simpler formulation of the model error covariance matrix. The STOI method improves short-term forecasts by combining a free run forecast with past observations. The analysis makes use of the observed radial current components instead of 2D current vectors, which enables the analysis even if two of the three antenna stations fail. In its present implementation, the analysis is based on the assumption that the model errors are dominated by inaccuracies in the timing and amplitude of the tidal wave. In this case, the model error covariance matrix has the same correlation structure as the model background
covariance matrix and can thus be estimated from a longer model run. To apply the STOI method, the covariance structure of several subsequent time steps has to be estimated. For this purpose, an EOF analysis of the state vectors comprising 24 consecutive hourly time steps was performed using model data covering a period of 3 months. More than 95% of the variance can be explained by the first 6 modes. The other important component in the analysis equation is the observation error. The HF radar data contain information about measurement accuracies, which are based on the properties of the
respective Doppler spectra. The HF radar data are quality controlled before inclusion in the analysis. The quality checks ensure that unrealistic current velocities and temporal velocity changes are disregarded. The observation accuracy maps, together with the model EOFs, are the basis for the evaluation of the Kalman analysis equation covering 24 time steps. When the observation error is large compared to the model error, the analysis tends to stay close to the free model run. If the quality of the observation data is high, the analysis attempts to reduce the deviation between the free model run and the

measurements. In forecast mode, observations are only used during the first part of the analysis window (e.g., 18 hours), and the analysis provides a corrected forecast for the remaining period (e.g., 6 hours). The advantage of this method compared to the classic filter approaches is that a smooth current field evolution is obtained over the entire period covered by the analysis window. Because the chosen length of the analysis window covers at least one period of the dominant M2 tide, the method is

efficient in correcting the phase errors of the tidal wave. More details of the approach are described in Stanev et al. (2015b). The STOI method was implemented as part of the pre-operational COSYNA system. The output is freely available from www.cosyna.de. An example of an analyzed current field, together with the free model run and the HF radar measurements, is shown in Fig. 9b. The plot shows the outflow situation on 26.03.2012 at 03:00 UTC, with current speeds approaching 1 m/s in some areas. The high variability of the currents is mostly due to the combination of strong tidal forcing and specific

bathymetric features in the very shallow area. The green arrows represent the HF radar measurement, the blue arrows are the free model run and the red arrows are the analysis.

Table 4 shows the innovation and analysis residuals for the radial components of all three HF radars averaged over a period of 3 months. Innovation is defined as the RMS difference between the observations and free model run. The analysis residual is the RMS difference between the observations and the analysis. The percentage reduction (RED) shows that the STOI

scheme achieves an improved agreement with the HF radar observations with regard to all measured velocity components.

The skill of the data assimilation estimated using independent ADCP data is demonstrated in Table 5. The RMS differences with respect to the ADCP measurements taken at the FINO-3 platform (see Fig. 9b for its location) are presented, with the first and second column referring to the free model and the analysis and the last column representing the achieved reduction in percent. The RMS values represent the averages computed for a period of 3 months. The reduction values demonstrate

that the analysis improves the surface current estimates compared to independent measurements, which were not used in the assimilation procedure.

The positive outcome of the data assimilation extends beyond the HF radar covered area, i.e., the analysis scheme has upscaling capability (see Stanev et al., 2015b and section 3.4, where the upscaling problem is addressed in more detail). The analysis scheme can be run both in hindcast and forecast mode. The above demonstration of skill is valid for the specific

area. Further application of the proposed method to different regions (e.g., regions dominated by pronounced baroclinicity) requires additional analysis.

### 3.2.2 Particle tracking: Enhancement of search and rescue using COSYNA surface currents

Maritime safety, marine resources management, coastal and marine environment protection, coastal weather forecasting, and monitoring and seasonal and longer-term forecasting of the coastal climate require the integration of existing and newly

emerging technologies to provide society with the best estimates, including the quantitative information about error. Accurate predictions in tidal-dominated environments have large practical value because small errors in tidal phase can largely impact the success of search and rescue operations. The use of COSYNA surface currents can improve short-term forecasts, which is of practical relevance, e.g., for search and rescue.

The following two experiments are conducted using the model data described in section 3.2.1 as the input (see also Fig. 9). In total, 33,746 Lagrangian particles (the number of wet model points) are released every day starting from 00:00 on 01.09.2011 at the surface in the center of every grid cell and are 2-D tracked with a Lagrangian model. The trajectories are computed during three days using the hourly model output from either the analysis or free model run. The trajectory simulations for the same initial positions of particles are restarted every day for the same integration time for three days. The Lagrangian model output consists of 33,746x30x24 individual positions. In Fig. 10a, the monthly averaged distance between positions of particles in the two runs 24 hours after release is shown. Release locations from which particles reached the model boundary are excluded from the statistical analysis.

This map provides an idea about the expected success of search and rescue if data from the HF radar are used or not used. In the latter case, the positioning of a lost object would be incorrect by 3-6 km after one day. Errors could be particularly large if the release is in the proximity of barrier islands or close to the northern model boundary. The complicated mesoscale currents around Helgoland Island pose problems in the model and observations and explain the larger spatial variability of the error pattern. The trajectories from the two runs in 6 locations during three days of integration starting on 05.09.2011, which are shown in the same figure, provide information about the dominating propagation patterns and indicate that the coherence of tidal oscillations is lost relatively soon after release. This illustrates the need for intra-tidal information from measurements to correct model trajectories.

The temporal evolution of the distance between particles released at the same position (Fig. 10b) demonstrates the rapid increase in the distance between trajectories in the two runs. The averaged positioning error plotted by the dashed line gives an overall idea about the accuracy of search and rescue operations using output from the free run. The reduction of the error of the positioning of an object due to the use of HF radar data during three days is approximately 10 km on average. The HF radar data enhance the quality of the surface current products in search and rescue applications.

### 3.3 Assimilation of the surface temperature and salinity data

SST is one of the fundamental parameters affecting water, heat and momentum exchange with the atmosphere, making these data a valuable component for assimilation in ocean modelling. SST data can be obtained both *in situ* and by remote sensing and can be derived in the German Bight as a comprehensive surface temperature analysis from remote sensing and as point observations from stationary and mobile platforms. One example of mobile platforms is the Ferry Box system, which is an autonomous measurement, data logging and transmission system, in which temperature and salinity measurements from samples of a continuous flow of seawater are taken from a water depth of 4-6 m while the carrying ship is traveling (Petersen et al., 2007). As shown by Grayek et al. (2011), the Ferry Box can enhance the SST hindcast in the German Bight near the Ferry Box track.

The use of remote sensing analysis data from the Operational Sea Surface Temperature and Sea Ice Analysis (OSTIA, Donlon et al., 2012) is an alternative to using Ferry Box data. The advantage of the OSTIA data is its spatial coverage; however, due to the applied processing procedures, the data are smoother than other SST products (Fig. 11a, b, see also,

Donlon et al., 2012). Furthermore, errors in the OSTIA data are larger in the coastal zone than in the open sea (see Fig. 15 of Stanev et al., 2011). To provide an optimal analysis with sufficiently high resolution, an SST assimilation routine is used to address all possible temporal/spatial combinations of the data sources mentioned above, allowing inclusion or exclusion of a portion of the observations.

The numerical model used to test the impact of different data on the performance of the data assimilation system is the one-way nested German Bight model based on GETM, which was presented in the previous sections (Staneva et al., 2009). The sea surface temperature and salinity data assimilation system follows a Kalman Filter approach. The method makes use of a priori information about the background statistics, which is estimated from a free run (see for more detail Grayek et al., 2011). For the daily analysis, the measurements derived from the different observation platforms are mapped on the model

grid and blended with the model forecast. Temporal correlations are not taken into account during the analysis. The yearly averaged RMS difference between the free run and the OSTIA data does not exceed 0.5°C over a large section of the German Bight (Fig. 11c). The overall estimate of the skill of the data assimilation is demonstrated in the following validation against independent observations at the Ems Station (see Fig. 1 for its location). The temporal evolution of errors and gain estimated for the same data station are shown in Fig. 11e. These results prove that assimilating the OSTIA data reduces the

errors substantially. The errors in the Wadden Sea, which remain relatively large, are attributed to the coarse model resolution (1 km), as well as to possible problems with the relatively coarse OSTIA data. Although the OSTIA patterns are correct overall, large differences exist in the coastal zone between these data (Fig. 11a) and the data in Fig. 11b, originating from the L3 multi-sensor super-collated SST product (CMEMS 2015). This motivates a closer look (using higher resolution) at the simulations of the near-coastal zone and estuaries (see section 2.5).

The assimilation of OSTIA data complemented by SST and SSS observations from the Ferry Box system also improves the skill of the model (earlier estimates of the skill in a similar assimilation experiment were presented by Grayek et al., 2011 and Stanev et al., 2011). The added value of assimilating Ferry Box data is clearly demonstrated in the time series of the surface temperature during 2011 (Fig. 12a) and in the time series of the surface salinity (Fig. 12b) when compared with the observations from the Ems Station. The RMS difference between the OSTIA data and observations is 0.46°C in 2011. The

difference between the free model run and observations, which is 1.05°C, is reduced to 0.36°C by assimilating the data. More interesting is the impact of data assimilation on the simulated salinity. In the free run, the RMS difference between observations and simulations in the Ems data is 0.73 psu. When only the SST from the Ferry Box is assimilated, the error is reduced to 0.66 psu, which means that the improved thermal state improves the salinity. Additional improvements from assimilating Ferry Box salinity are observed, with the RMS difference decreasing to 0.52 psu.

The L3 multi-sensor super-collated SST product (CMEMS 2015) provides higher spatial resolution (Fig. 11a) and more information on smaller scales, but its temporal/spatial coverage depends on the satellite's revisit time and the cloud conditions. In another analysis, these finer-resolution data were assimilated (not shown here). The comparison between the experiment assimilating the OSTIA data only and the experiment assimilating the super-collated SST product reveals that the latter improves the representation of the short-term and small-scale variability, which is associated with the synoptic-scale

features. However, similarly to the case with the regional NWP system, maximizing the use of observations in small model domains requires more frequent update cycles (deHaan, 2013). Without continuously using the L3 multi-sensor super-collated SST product (CMEMS 2015), the imprint in the model solution persists (in the strongly tidally driven coastal ocean) for only a couple of days. Unfortunately, such SST products are not always available because of changing cloud conditions.

The use of observations from some existing fixed platforms (e.g., platforms of opportunity, such as offshore wind farms) could provide an alternative to enhance the COSYNA pre-operational system by more accurately accounting for the high-frequency processes.

## 3.4 The concept of upscaling

An increasing number of coastal observatories are becoming operational worldwide (Riethmüller et al., 2009; Stanev et al.,
2011; Howarth and Palmer, 2011; Baschek et al., 2016). This development is driven by the growing need for information about coastal processes that is relevant to the planning and management of human activities, e.g., offshore wind farming. At the same time, major efforts are underway in different parts of the world to develop operational models for the regional scale. For example, in Europe, these activities are organized in the framework of the Copernicus Marine Service (http://www.copernicus.eu/), which ensures consistent regional model forecasts are provided for all European coastal areas.
Regional models, such as the North West Shelf model used in the Copernicus Marine Service (O'Dea et al. 2012), cannot resolve all relevant coastal processes. Downstream services for user groups interested in coastal information usually require higher spatial resolution. The usual approach to solve this problem is a nested setup, where a high-resolution coastal model is coupled to a coarser model ("parent model") using either one-way or two-way coupling methods (Barth et al., 2005). Alternatively, unstructured-grid models are used to achieve a seamless transition between different spatial scales (Zhang et
al., 2016b). Due to the high computational costs, the use of these models for operational applications is currently limited. Additionally, the assimilation of observation data usually requires the use of high-resolution models because many small-scale processes, e.g., those monitored by HF radar systems, cannot be reproduced by regional-scale models. To make best use of coastal observations and to improve both coastal- and regional-scale forecasts, different aspects of nested model coupling require detailed analysis.
While the propagation of information from coarser regional models to high-resolution coastal models ("downscaling") is straightforward and well established, the information flow in the opposite direction ("upscaling") is demanding and is the subject of ongoing research. Different aspects of this problem are discussed in Schulz-Stellenfleth and Stanev (2016) for the barotropic dynamics in the North Sea. The two main aspects are: 1) the impact of small-scale information on the regional scale at the same location and 2) the impact of small-scale information at one place on the regional scale at another place.
The first aspect is relevant in the context of parameterization errors (e.g., bottom roughness), while the second aspect is important for the assimilation of coastal observations (which was illustrated in section 2.3 when describing the coupling between the North Sea and Baltic Sea).

The analysis of the upscaling problem presented in Schulz-Stellenfleth and Stanev (2016) is based on a 2D linear barotropic model for the North Sea, which can be extended to include nonlinear bottom friction effects. The model solves the Navier–Stokes equation in the spectral domain using complex coefficients describing the phase and amplitude of the water elevation and current components. From the numerical point of view, model solutions are obtained by solving a large banded complex linear system of equations. The advantage of this approach is that the model equations can be inverted in a relatively straightforward way.

The potential of coastal observations to improve boundary forcing on a larger scale was demonstrated in Barth et al. (2010). The method proposed by Schulz-Stellenfleth and Stanev (2016) makes it possible to estimate, in a generic way, the model response statistics for a given model parameter perturbation covariance. This can also be used to estimate the influence of the coastal boundary condition on the performance of the North Sea model. The spectral model of Schulz-Stellenfleth and Stanev (2016) is implemented in a two-way nested configuration with a 5 km North Sea grid and a 1 km grid for the German Bight. The impact of perturbations from the 1 km model on the North Sea model is analyzed by introducing the German Bight boundaries as new open boundaries in the larger model. One can interpret these perturbations as small-scale corrections resulting from observations inside the German Bight, e.g., caused by assimilating HR radar observations in the German Bight area. The example in Fig. 13 shows the water level standard deviation resulting from perturbations along the German Bight boundary. If these observations are correlated, as is the case in this example (e.g., adjustments of the timing or amplitude of the tidal wave inside the German Bight), they have a strong impact not only in the vicinity of the open boundary but also at the English coast, the latter being almost as large as the original perturbation at the German Bight boundary. This demonstrates that observations acquired in the German Bight have a significant impact on the North Sea scale with the exact impact characteristics depending on the setup, e.g., the open boundary conditions used for the North Sea model.

## 4 Conclusions

One of the aims of the present study was to present recent developments in coastal ocean forecasting with a focus on new modelling issues, coupling between models, data assimilation and research focused on practical applications. The large number of issues and the fact that some previous publications addressed the individual research question led us to the decision to review the modelling-related COSYNA activities, some of which have been already implemented in the COSYNA pre-operational suite.

Tidal forcing is at the heart of the North Sea dynamics, and it has been given much consideration in the past. However, there are insufficient inter-comparison studies on the performance of individual models to adequately simulate tides, particularly shallow-water tides. These shallow-water tides are very important for the tidal dynamics in coastal areas, where the tidal range is larger than in the open ocean and the propagation of tidal waves is more complex. One manifestation of this complexity is nonlinear physics, which many models do not accurately simulate. This is illustrated by the comparison

between the simulated shallow-water tides in the individual models. We compared the tidal analyses from six model setups for almost the same ocean area and demonstrated that the M4-tide amplitudes show pronounced differences in all models. Unlike the good agreement in the M2 phases, the phases of the shallow-water tides differ substantially, which is just one demonstration of how different the representation of the nonlinearity in the analyzed regional models is. With the exception

of SCHISM, the models used horizontal resolutions of several kilometers, which is not sufficient to resolve shallow-water tides. This is one explanation of the differences in the simulated nonlinear tidal dynamics, particularly the distortion of the tidal signals in the shallow coastal zone of the North Sea.

In all models, the M4 tidal patterns have smaller spatial scales than the M2 tides. Although they have very small magnitudes, these oscillations are very important for the asymmetry in the tidal dynamics of coastal ocean and the resulting net transport

of matter. Therefore, further attention is needed with respect to the adequate representation of shallow-water tides in the numerical regional and coastal models.

The model inter-comparison demonstrated that the closed boundary in the Danish Straits in the FOAM-AMM7, which is set up only for the north-west shelf, results in the generation of an amphidromic point in the Kattegat, which was not observed in the other models (coupled with the Baltic Sea). This issue can easily be solved, as shown by most of the model

simulations described here, by including the Baltic Sea. However, the optimal nesting to be applied to the transition area between the North Sea and Baltic Sea is still unknown. We demonstrated the performance of a new two-way nesting method enabling the use of different vertical discretization in the individual nests of the North Sea–Baltic Sea NEMO using regionally appropriate vertical resolution (σ-coordinates in the shallow and tidally dominated North Sea and z-coordinates in the strongly stratified Baltic Sea) to avoid problems with the adequacy of application to different ocean areas (in the

illustrated case, the shallow and well-mixed North Sea and the strongly stratified Baltic Sea). The presented method allows upscaling of information from the fine-resolution nest to the coarser-resolution simulation while maintaining the overall dynamic consistency. Fine-resolution nesting enhances the long-channel changes in stratification, which can be explained by the more realistic simulation of the secondary (transversal) circulation.

The use of unstructured-grid models is an unavoidable research path when addressing the coupling between different models

in the narrow straits. We did not consider this issue for the Baltic Sea straits, although the numerical simulations with SCHISM were included in the inter-comparison in section 2.2. This is because this issue was addressed in detail by Zhang et al. (2016a), and we decided to illustrate a different application of unstructured-grid modelling. Using SCHISM as a tool to address the estuarine dynamics, we investigated the representation of tidal asymmetry in the Ems Estuary. The ratio between the amplitudes of the M4 and M2 tides reaches 0.3, indicating a strong distortion of the tidal wave. The transport resulting

from this tidal asymmetry could affect the accumulation of sediment and the transformation of fresh water from the tidal river to the open ocean. This transformation is characterized by a pronounced asymmetry in the horizontal plain. Spring-neap variability plays a major role in shaping the export pathways of fresh water from the estuary. Since the estuarine spatial scale cannot be resolved by the larger-scale models, development of the capabilities of unstructured-grid models to provide a

better representation of the coastal to open ocean exchange is of high priority. This would allow an appropriate description of the river boundary condition in the regional- and larger-scale models.

COSYNA provides a large amount of data from different platforms and sensors. Among these, HF radar data, with their large area coverage, high resolution in time and space, and long-term operational capability, could become a valuable data source for monitoring surface currents and other operational activities. We demonstrated the contribution of the developed spatiotemporal optimal interpolation (STOI) filter to improve the short-term hindcasts and forecasts of the surface currents using observations from three continuously operating radars in the German Bight. The method provides short-term forecasts of the surface currents and was implemented as part of the pre-operational COSYNA system. The analysis provides a continuous and homogenous data series over the entire German Bight and improves the agreement of the model with the radar observations. One application of the STOI to improving the quality of Lagrangian particle tracking showed that the reduction of the error of positioning of an object due to the use of HF radar data during three days is approximately 10 km on average, which indicates that the HF radar data can be used to enhance search and rescue missions.

The experiments assimilating temperature and salinity data showed that OSTIA data do not substantially improve the quality of predictions in the Wadden Sea. This is attributed to the coarse resolution of the model (1 km model resolution was used in these experiments), as well as possible problems with the relatively coarse-resolution data that were assimilated. The use of observations from the Ferry Box system improves the skill of the model. The assimilation of SST data with very high spatial resolution greatly improves the representation of the synoptic-scale features. However, the temporal and spatial coverage of these data depend on the satellite's revisit time and the cloud conditions. Without frequent use of these observations (because of missing data), their imprint on the model solution persists only for a couple of days because the "memory" of the shallow-water physical system is short.

Another issue that is relevant to the regional ocean forecasting is that models are not able to resolve all relevant coastal processes. The propagation of information from coarser regional models to high-resolution coastal models ("downscaling") is straightforward and well established. However, the information flow in the opposite direction ("upscaling") is demanding and is the subject of ongoing research. We provided one illustration of the impact of small-scale information at the coastal scale on the regional-scale motion. This elucidated the potential of coastal observations to improve the boundary forcing on a larger scale, which is crucial when coupling between coastal and regional forecasting systems is considered.

The improvement in the quality of the predictions of wind waves in the coastal ocean has large practical value. This improvement is strongly dependent on the model physics, which are very sensitive to shallow depths. We demonstrated that the large differences between the numerical simulations in coupled and un-coupled models indicate that accounting for the wind-wave effects in the three-dimensional hydrodynamic model improves predictions in the shallow coastal waters. The presented illustrations from coupled and un-coupled wave and circulation models indicated that a large part of the uncertainty results from the nonlinear interaction between strong tidal currents and wind waves. This factor cannot be ignored in the theoretical studies and operational applications, in particular in the coastal zone, where its role is dominant. Thus, precise coastal predictions cannot neglect the wind-wave–current interaction in the coastal ocean. In particular,

predictions of storm events with coupled models could be of utmost importance for many coastal applications involving risk analysis (e.g., offshore wind industry, oil platform operations.), where higher accuracy is needed.

Finally, some of the examples considered here are relevant to the service evolution strategy of the Copernicus Marine Environment Marine Service (CMEMS, see also http://copernicus.eu/) and can be considered as a research input for operational oceanography. The relevant document, which is prepared by the CMEMS Scientific and Technical Advisory Committee (STAC), available at (https://www.mercator-ocean.fr/wp-content/uploads/2015/11/13-CMEMS-Service_evolution_strategy_RD_priorities.pdf), has largely motivated us to report the results of our recent research.

## 5 Annexes

### 5.1 NEMO

The primitive equation ocean model NEMO (Nucleus for European Modelling of the Ocean) is a flexible tool for studying the ocean over a wide range of space and time scales (Madec, 2008). In the inter-comparison study considered in section 2, model data from two NEMO setups are used. The first data set is from the Copernicus Marine Environment Monitoring Service (http://marine.copernicus.eu) for the north-west shelf. The output is provided by the Forecasting Ocean Assimilation Model-Atlantic Margin model with 7 km resolution (FOAM-AMM7, O'Dea et al. 2012). The model uses 32 terrain-following sigma-levels in the vertical direction.

The second setup is for the North Sea–Baltic Sea at the Helmholtz-Zentrum Geesthacht (NEMO-HZG). The model area for the second setup extends from -4°9′E 48°29′N to 30°11′E 65°54′N. In its standard version, the model uses 2 nm resolution in the horizontal direction and 21 sigma-levels in the vertical direction. The model area is shown in Fig. 2.

The air-sea fluxes are estimated using bulk formulas and 6-hourly atmospheric forcing fields derived from the ERA-Interim hindcast data set, which is produced by the European Centre for Medium-Range Weather Forecasts (ECMWF). At the open lateral boundaries, the model is forced with 4D interpolated temperature and salinity profiles calculated from the monthly climatological data of Janssen et al. (1999). The boundary conditions for the currents and sea-surface elevation at the lateral boundaries are derived from harmonic tidal analyses provided by Oregon State University Tidal Inversion Software (OTIS). Water fluxes at the surface include ERA-Interim precipitation.

### 5.2 GETM

GETM (General Estuarine Transport Model) is a primitive equation prognostic three-dimensional hydrodynamic model. The use of generalized vertical coordinates makes it suitable for shallow coastal regions under the influence of tidal currents (Burchard and Bolding, 2002). In this model, the equations for the three velocity components, sea-surface height, temperature, salinity, turbulent kinetic energy and the eddy dissipation rate due to viscosity are solved. A particular feature of GETM is its ability to adequately represent the dynamics in deep inlets and channels and on the tidal flats, the latter falling dry during part of the tidal period (Stanev et al., 2003).

The nested modelling system based on GETM consists of three model configurations: a coarse-resolution (approximately 5 km) North Sea–Baltic Sea outer model, a fine-resolution (approximately 0.8 km) inner model covering the German Bight and a very fine-resolution (approximately 200 m) model for the Wadden Sea region resolving the barrier islands and the tidal flats (for more detailed presentation of the model area and set up, see Staneva et al., 2009). The bathymetric data for the different model configurations are prepared using the ETOPO-1 topography, together with observations made available from the BSH. The model system is forced by: (1) atmospheric fluxes estimated by the bulk formulation using hourly forecasts from the German Weather Service (DWD), (2) hourly river runoff data provided by the BSH operational model, and (3) time-varying boundary conditions of sea-surface elevations and salinity. The sea-surface elevations at the open boundary of the North Sea–Baltic Sea model are generated using tidal constituents obtained from the TOPEX/POSEIDON harmonic tide analysis. The temperature and salinity at the open boundary of the outer model are interpolated at each time step using the monthly mean climatological data of Janssen et al. (1999). The fresh water fluxes from the main tributaries in the region are taken from the observations available from the Niedersächsischer Landesbetrieb für Wasserwirtschaft und Küstenschutz, Aurich, Germany.

### 5.3 BSHcmod

BSHcmod is a three-dimensional prognostic model (Dick et al., 2001; Dick and Kleine, 2007) that was developed at the German Federal Maritime and Hydrographic Agency (BSH). The model is used for operational applications and is run in a two-way nested configuration to provide information on the sea level, temperature, salinity and currents in the North Sea and the Baltic Sea. In the standard operational setup, the model is used for 72 hr forecasts. The model provides input to drift and dispersion models, which are performed on demand.

A version of the model described above with adaptive vertical coordinates (Dick and Kleine, 2007) was made available to the authors by the BSH and was used to generate an additional output data set that was analyzed in the present study. A grid with 900 m resolution was used for the German Bight, and a coarser resolution of 5 km was used for the remaining portion of the North Sea and Baltic Sea. The open boundary conditions are formulated using sea-level data calculated from the tidal constituents of 14 partial tides. Data from the German Weather Service (DWD) were used for the wind forcing, and climatological data were used for river discharge. The code was run on 8 processors using the OpenMP library. The data analysis is based on model results for the year 2011. The 5 km model grid is identical to that of the operational model described above (see Fig. 2).

### 5.4 SCHISM

Unstructured-grid models enable a seamless transition between processes at coastal and open-ocean scales. SCHISM (Semi-implicit Cross-scale Hydroscience Integrated System Model; Zhang et al. (2016b)), which is a successor of the original model of Zhang and Baptista (2008), is one such model. New developments since the last publication (Zhang et al. 2015) include the addition of a mixed triangle-quadrangles grid and 1D/2D/3D options wrapped in a single model grid. SCHISM

solves the hydrostatic Reynolds-averaged Navier–Stokes equations with the transport of heat, salt and tracers in the hydrostatic form with Boussinesq approximation on unstructured grids. The efficiency and robustness of SCHISM are mostly attributed to the implicit treatment of all terms that place stringent stability constraints (e.g., CFL) and the use of the Eulerian-Lagrangian method for the momentum advection. The vertical grid allows the use of partial terrain-following *S*-

and partial *z*-coordinates (flexible localized sigma coordinates with shaved cell, Zhang et al., 2015).

The setup of SCHISM for the North Sea–Baltic Sea area is described by Zhang et al. (2016a); the model area is shown in Fig. 2. Altogether there are ~300K nodes and ~600K triangles, with refinement along the German Bight and Danish straits, where a nominal resolution of 200 m is used. On the open North Sea boundaries (Scottish Shelf and English Channel), time series of the elevation, horizontal velocity, salinity and temperature are interpolated from the MyOcean products

(http://www.myocean.eu).The sea-surface boundary conditions use the output from the regional model COSMO-EU (wind, atmospheric pressure, air temperature and specific humidity) operated by the German Weather Service with a horizontal resolution of 7 km. Heat fluxes (including solar radiation and downward long-wave (infrared) radiation), which are needed as a surface boundary condition, come from the NOAA's CFSR product (http://www.ncdc.noaa.gov/data-access/model-data/model-datasets/climate-forecast-system-version2-cfsv2). Monthly flow data at 33 rivers in the region are provided by

the BSH.

### Acknowledgements

This study combines research conducted in the frame of the German Helmholtz PACES (Polar Regions And Coasts in the changing Earth System) program and the Earth System Knowledge Platform (ESKP). The applications for the Ems Estuary are supported by the BMBF-funded Future Ems project. We benefited from the data provided by COSYNA, BSH, DWD,

and the Copernicus Marine Service. We thank A. Barth for the helpful cooperation and Y. J. Zhang and B. Jacob for the support in the field of unstructured-grid modelling. BSH provided the BSHcmod code. Thanks are also due to I. Noehren who prepared some of the figures. The authors gratefully acknowledge the computing time granted by the John von Neumann Institute for Computing (NIC) and provided on the supercomputer JURECA at Jülich Supercomputing Centre (JSC).

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

**Figures and Tables.**

| Model | NEMO | | CMOD | | GETM | SCHISM |
|---|---|---|---|---|---|---|
| Setup | FOAM-AMM7 Version ? | NEMO-HZG Version ? | BSH-Operational | CMOD-HZG | | |
| Horizontal resolution | 7 km | 2 nm | 5 km | 5 km | 3 nm | > 60 m |
| Vertical resolution | 32 z*-σ-levels | 21 z-σ-levels | 36 z-σ-levels | 36 z-σ-levels | 21 σ-levels | max. 59, average 29 LSC²-levels |
| Forcing atmosphere | Met Office NWP model 3-hourly heat and moisture fluxes and 1-hourly wind and pressure, ~25 km | ECMWF ERA-INTERIM 6 hourly, ~14km | COSMO EU hourly, ~7km | COSMO EU hourly, ~7km | COSMO EU hourly, ~7km | COSMO EU hourly, ~7km |
| Forcing open boundary | 1-W-N into FOAM-AMM12 | Jansen et al. (1999) | 1-W-N into 2D10km CMOD-Modell | Jansen et al. (1999) | Jansen et al. (1999) | 1-W-N into FOAM-AMM7 |
| Forcing tides | 15 harmonic tidal constituents + one-way nested into FOAM-AMM12 | 9 harmonic tidal Constituents (OTIS) | 14 harmonic tidal constituents | 14 harmonic tidal constituents + 2D 10km CMOD-Modell | FES 2014 Tidal-Model-Solution | one-way nested into FOAM-AMM7 |
| Forcing rivers | climatological | climatological | E-HYPE (SMHI) | climatological | climatological | climatological |

Table 1. Short presentation of major characteristics of used models (see for details Annex)

Abbreviations: Numerical Weather Prediction (NWP), one-way nested (1-W-N),

| | RMSE | | BIAS (Model-Observations) | |
|---|---|---|---|---|
| | Coupled model | Circulation only model | Coupled model | Circulation only model |
| 01.10.2013-21.12.2013 | 9.4 | 14.4 | -3.6 | -9.5 |
| 05.12.2013-06.12.2013 | 12.1 | 25.3 | -8.7 | -17.4 |
| 07.12.2013-08.12.2013 | 10.7 | 16.2 | -5.6 | -11.4 |

Table 2. Performance of circulation-only and coupled wave-circulation models in reproducing sea level (in cm) against observations from 38 tidal gauges in the model area. Their positions are shown in the horizontal map of Fig. 6.

| Buoy | Number of comparisons | Mean of measurements | Bias | Root mean square error | Skill | Scatter index |
|---|---|---|---|---|---|---|
| $H_s$ | - | (m) | (m) | (m) | - | (%) |
| | | | | | | |
| old wave model version without wave breaking and with default wave growth parametrization | | | | | | |
| | | | | | | |
| Helgoland | 247 | 1.44 | 0.17 | 0.47 | 0.79 | 30 |
| Elbe | 87 | 1.70 | 0.25 | 0.47 | 0.78 | 24 |
| Westerland | 248 | 1.15 | 0.35 | 0.62 | 0.64 | 44 |
| | | | | | | |
| improved wave breaking and modified wave growth in the wind input source term | | | | | | |
| | | | | | | |
| Helgoland | 247 | 1.44 | 0.01 | 0.34 | 0.85 | 24 |
| Elbe | 87 | 1.70 | 0.08 | 0.35 | 0.83 | 20 |
| Westerland | 248 | 1.15 | 0.19 | 0.34 | 0.82 | 25 |
| | | | | | | |
| skill : reduction of variance, scatter index : standard deviation*100/mean of the measurements | | | | | | |

5     Table 3. $H_s$ statistics during storm Xavier in the German Bight. The positions of buoys are shown in Fig. 1.

| Radar Station | IN [m/s] | AR [m/s] | RED [%] |
|---|---|---|---|
| Sylt | 0.098 | 0.068 | 31 |
| Büsum | 0.176 | 0.139 | 21 |
| Wangerooge | 0.168 | 0.126 | 25 |

Table 4: Spatially and temporally averaged innovations (IN; m/s), analysis residuals (AR; m/s), and percentage reduction (RED) in the radial component differences after analysis with respect to the three radar stations.

|  | Free Run | Analysis | RED [%] |
|---|---|---|---|
| RMS u [m/s] | 0.122 | 0.097 | 20 |
| RMS v [m/s] | 0.126 | 0.103 | 18 |

Table 5: Comparison of the meridional and zonal surface current component of free run and analysis with respect to FINO-3 ADCP measurements (m/s). The last column is the achieved reduction (RED) in percent.

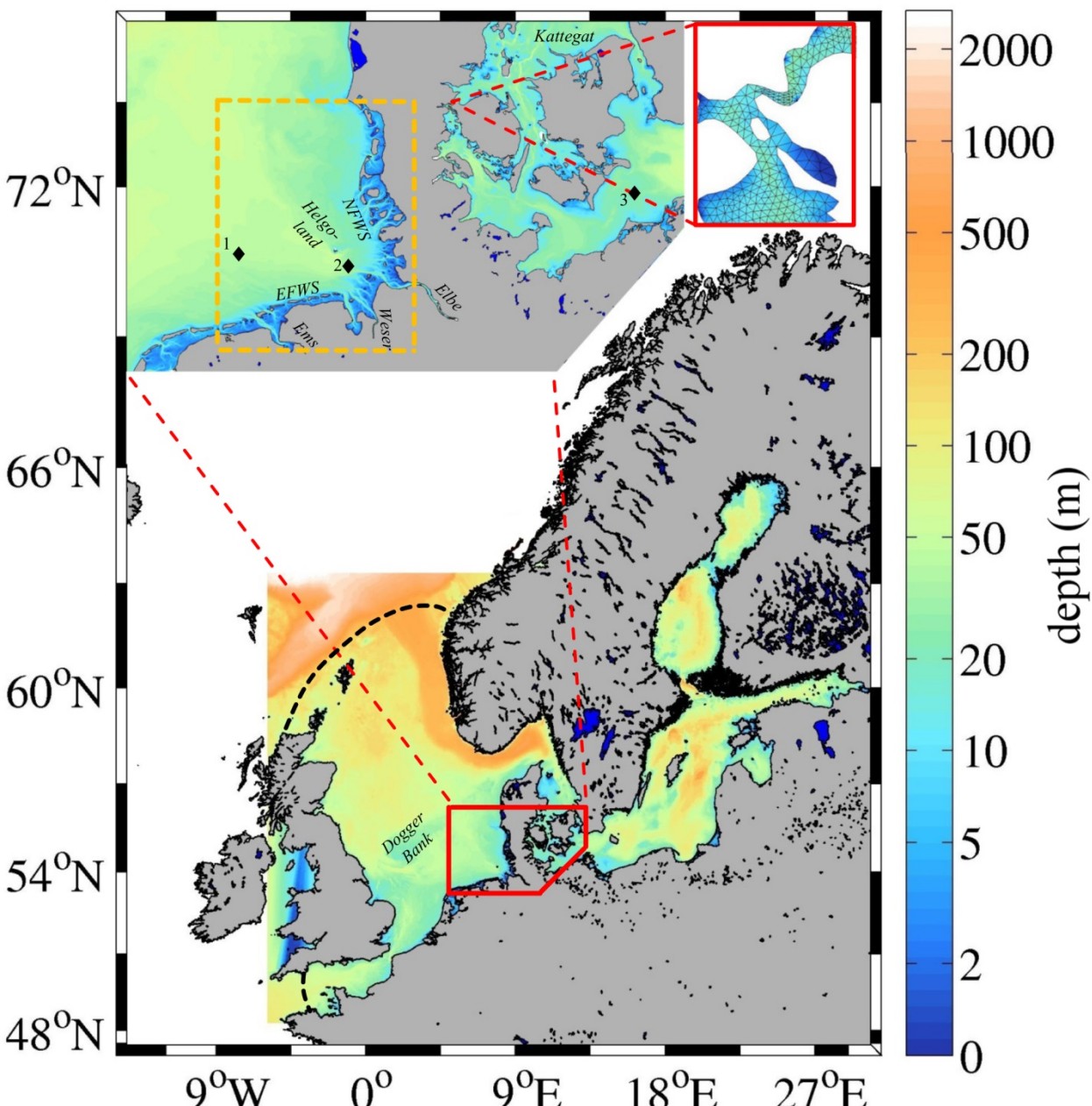

Figure 1: Bathymetry of the North and Baltic Seas. The zoom in the upper-left corner shows the bathymetric details of the German Bight and of the straits connecting the Baltic to the North Sea. The model area of the German Bight (COSYNA pre-operational model area) is shown in the upper-left zoom with the orange dashed line. The smaller zoom in the upper-right corner illustrates how fine the resolution needs to be in order to resolve the straits (in this case the Little Belt). The rhombi identify the locations of the stations (1 – Ems, 2 – Elbe, 3 – Darss Sill), which are part of the Marine Environmental Monitoring Network in the North Sea and Baltic Sea (MARNET).

.

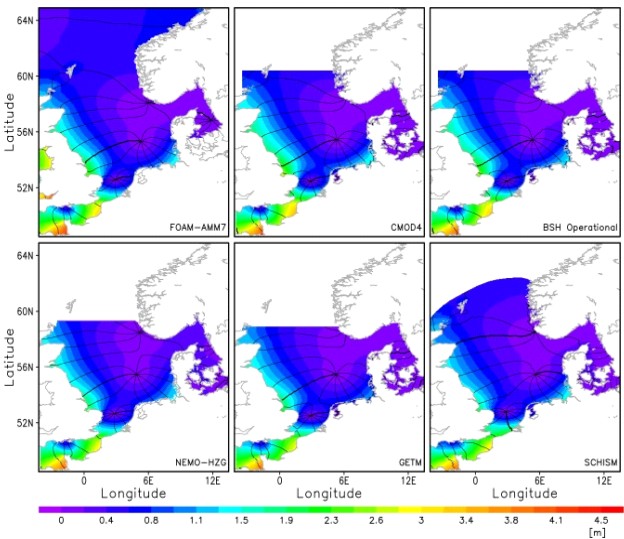

a)

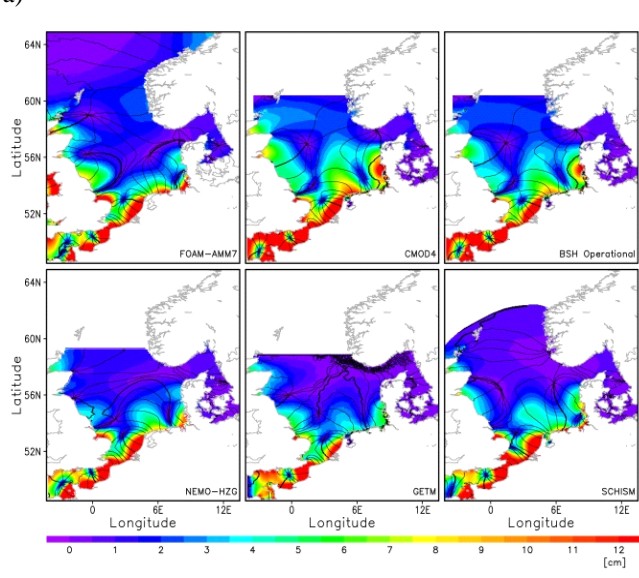

b)

Figure 2: Simulated M2 (a) and M4 (b) tidal amplitudes and phases from six different models operating in the North Sea. The simulations using NEMO-HZG, GETM, SCHISM and BSHcmod are carried out at the HZG. Estimates for FOAM-AMM7 and BSHcmod (operational) are derived from the freely-available data provided by the marine forecast services (Met-Office) and BSH, respectively.

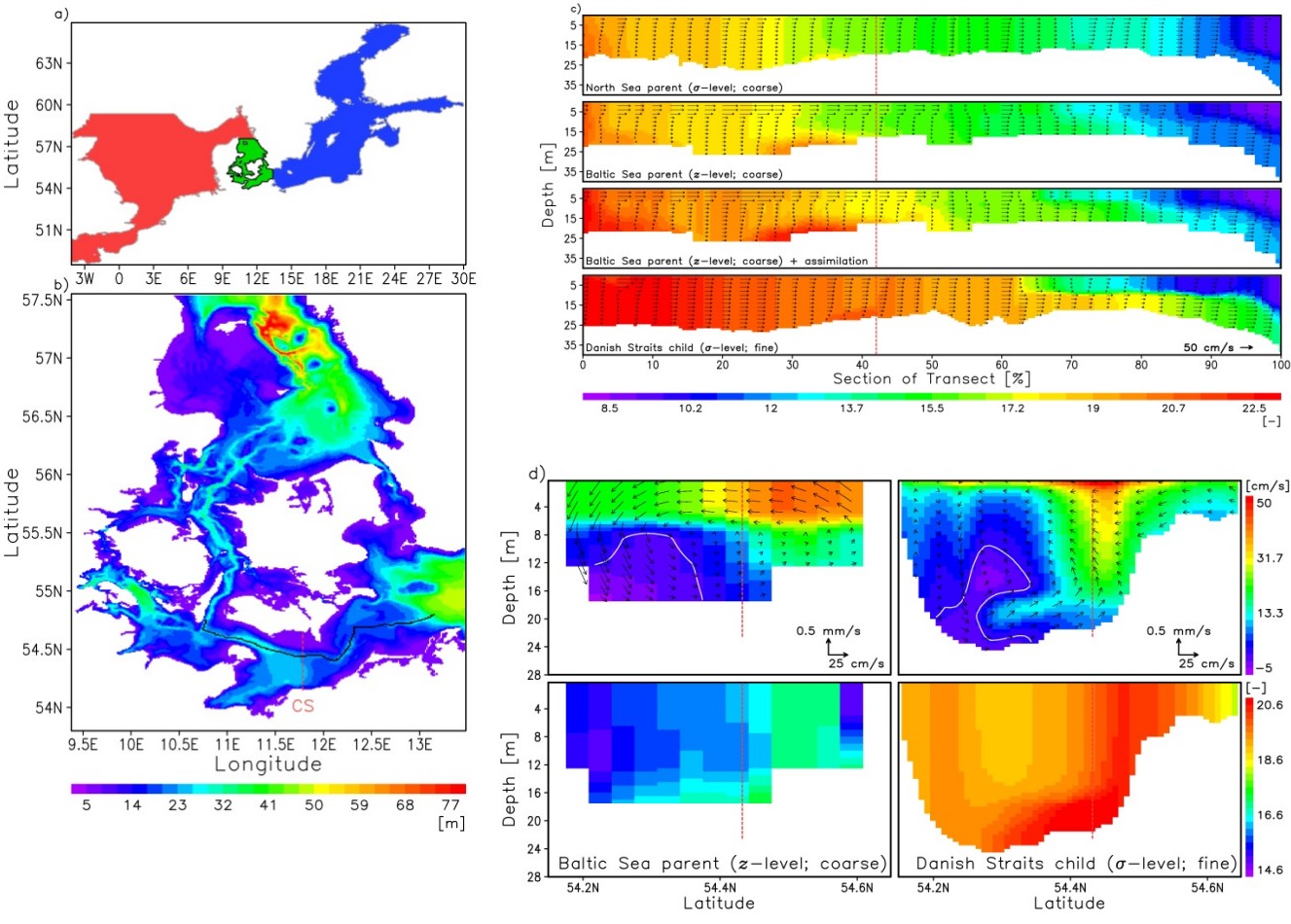

Figure 3. (a) Individual model areas: red+green+blue-colours depict the coarse-resolution model area for the entire domain; red+green and green+blue are the areas for the coarse-resolution North Sea and Baltic Sea models, respectively. The coupling between the different models is performed over the green area, which is presented in (b) along with the bathymetry as seen in the fine-resolution nested model. Sections along which some results are analysed (c and d) are also shown. (c) Snapshots of velocities (vectors) and salinity (colours) from the different model nesting experiments (names are given in the individual panels) during an inflow period on 16.09.2010. Data is shown for the long-channel transect (black line). (d) Across-channel section of velocity and salinity. The position of section is shown with the red line in (a).The position where the two sections cross is indicated by the vertical red lines in c and d.

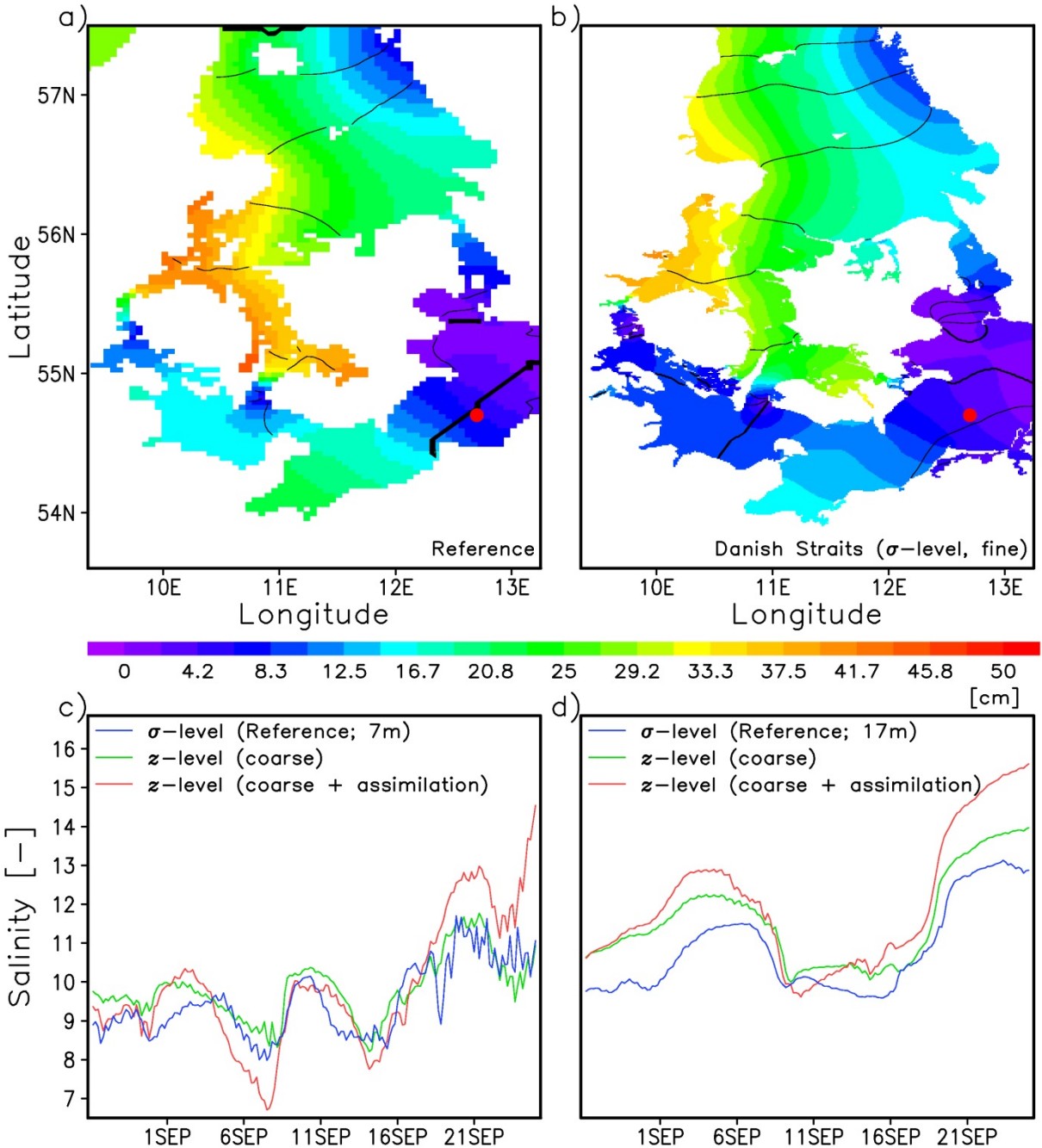

Figure 4. Tidal amplitude (in colours) and phase lines (the distance between individual lines is $40°$) in the transition zone simulated in the reference model (a) and the fine-resolution nest (b). (c) and (d) show the temporal change of salinity at 7m (c) and 17 (d) in a location in the Dars Sill, which is identified by the red symbol in (a, b). Legend explains the individual curves.

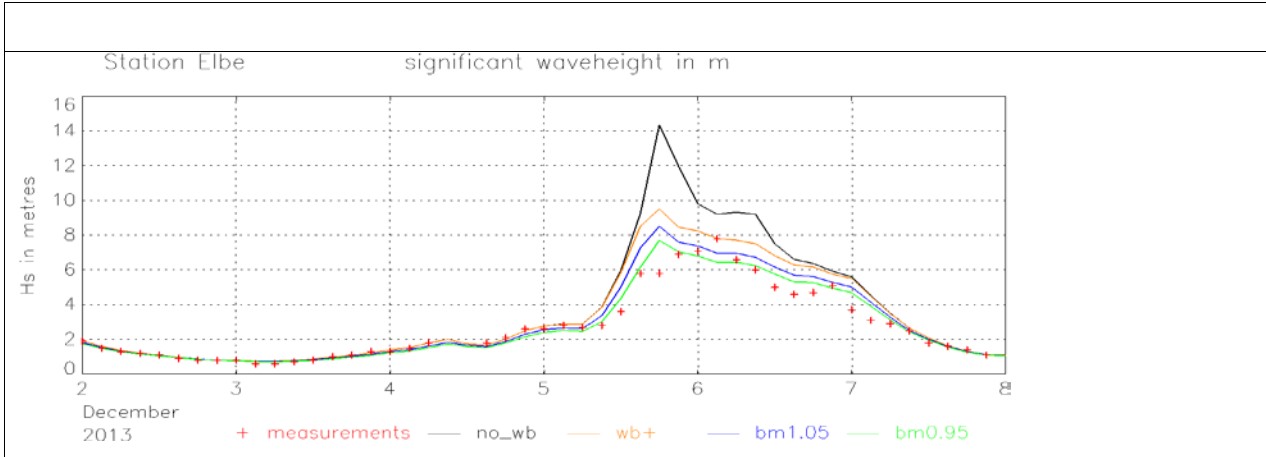

Figure 5. Time series of significant wave heights at Elbe station (see Fig. 1 for its position) during the storm Xavier (no_wb : run without wave breaking, wb+ : run with a new formulation for wave breaking, bm1.05 : run with a changed constant Betamax = 1.05 in the wave growth parametrization (default = 1.2), bm0.95 : run with Betamax = 0.95, the last two runs include wave breaking).

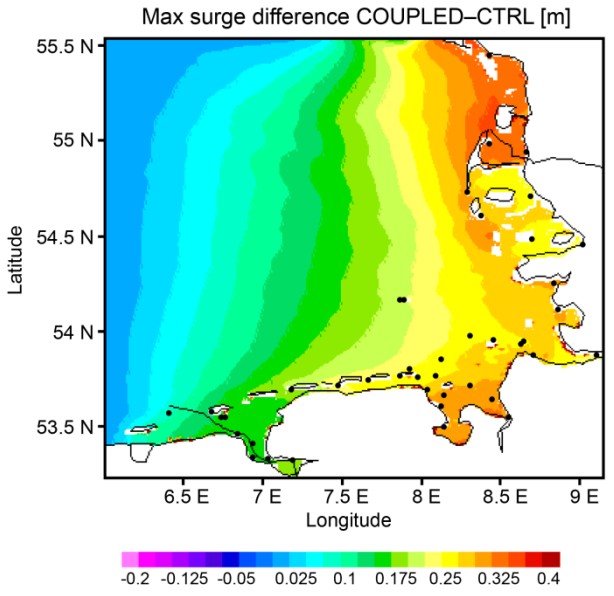

a)

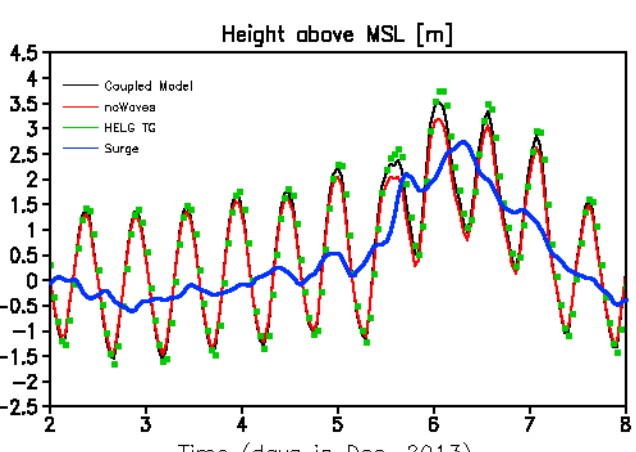

b)

Figure 6. (a) Difference between sea surface elevation (SLE) between coupled wave-circulation model (WAM-GETM) and the circulation- only model (GETM) for the German Bight during the storm Xaver on 06.12.2013. (b) Time series of Sea Level Elevation (SLE)  in [m] at Helgoland tide gauge station. Black line: coupled wave-circulation model; red line circulation only model; green:  tide gauge observations; blue line: the surge.

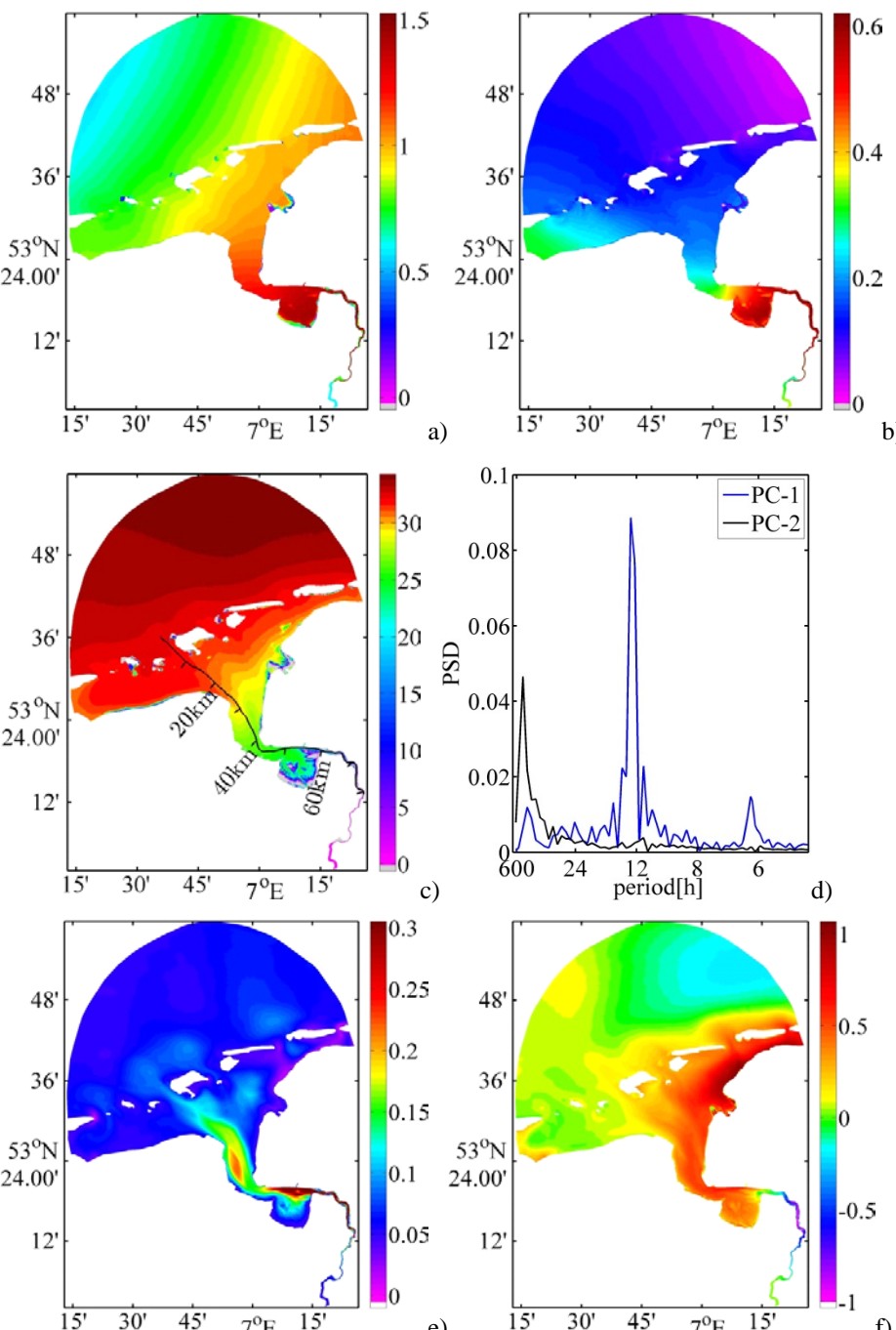

5     Figure 7. Amplitude of M2 (a) and M4 (b) tide in (m). (c) Vertically and time-averaged salinity [psu] during one neap-spring period (replotted from Pein et al., 2016). (d) Spectral analysis of the first and second PC. (e) First and (f) second EOF of surface salinity. Transect-line in (c) is where some analyses of model simulations are presented.

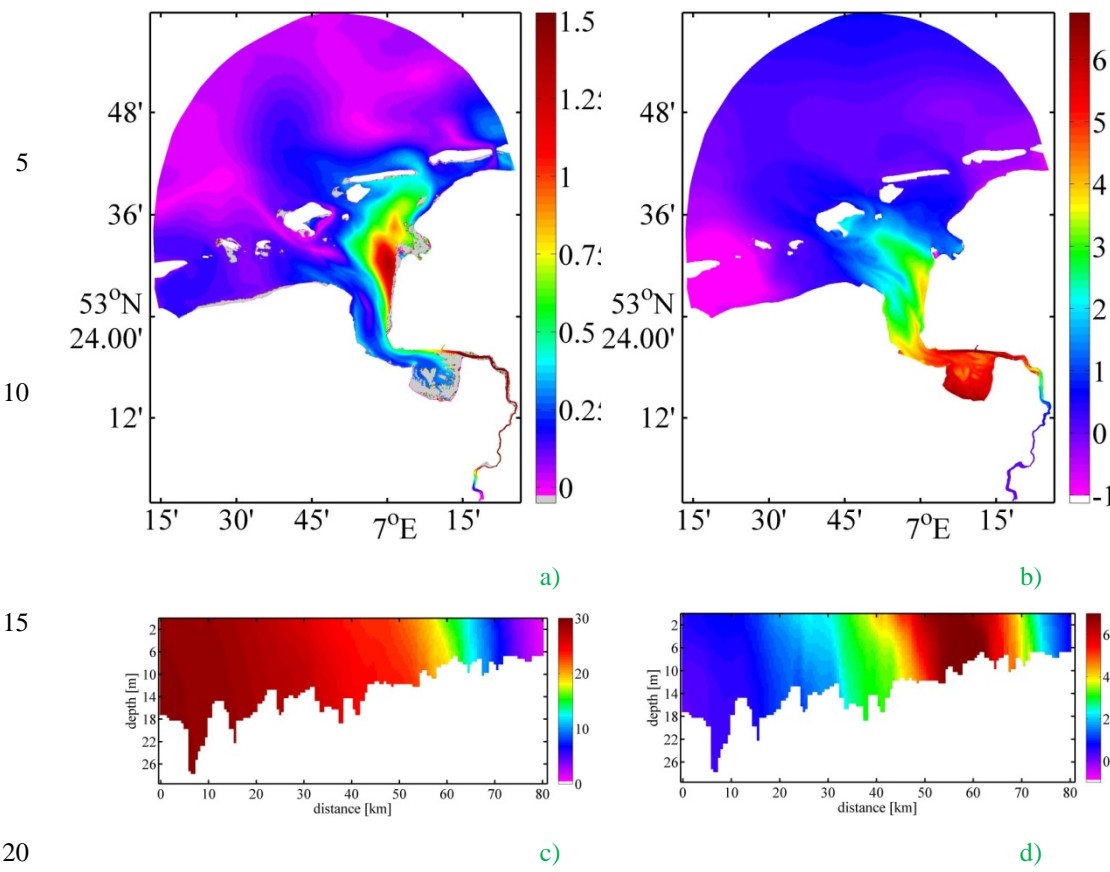

Figure 8. Dependence of surface salinity upon the neap-spring variability and the variability in river runoff. (a) Difference between simulated surface salinity during neap and spring tide. (b) Difference between salinity averaged over 8 tidal cycles (11.06.2012-15.06.2012, river runoff of 40 m^3/s) and 11.02.2013-15.02.2013, river runoff of 140 m^3/s. (c) Salinity in the low-river runoff case along the transect line shown in Fig. 7c. (d) corresponds to (b) but along the transect line.

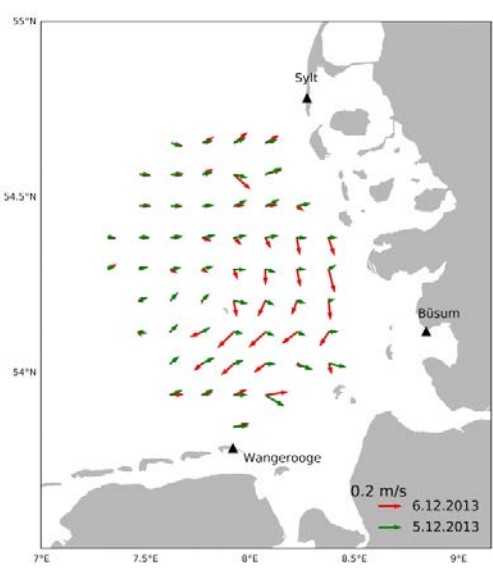

a)

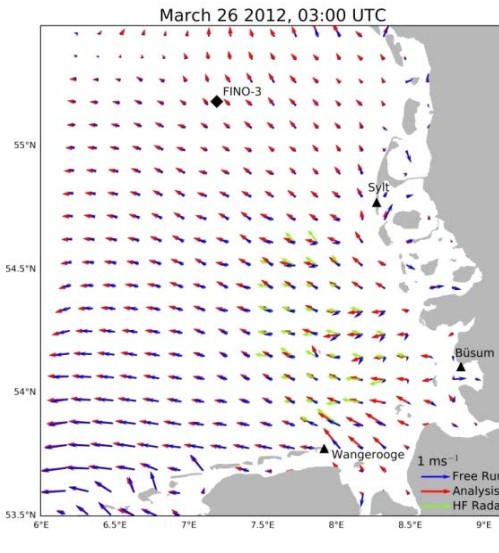

b

Figure 9. Surface currents in the German Bight. (a) De-tided currents as seen by the HF radars during the period when the storm "Xaver" was over the German Bight. The position of arrows illustrates the area covered by HF radar observations. This area is smaller than the model area. (b) Simulated and observed surface currents. The green arrows represent the HF radar measurement (not over the entire model area), the blue arrows are based on the free model run and the red arrows are the analysis. The positions of the three HF radar stations are given with the triangle-symbols, the diamond shows the position of FINO-3 station.

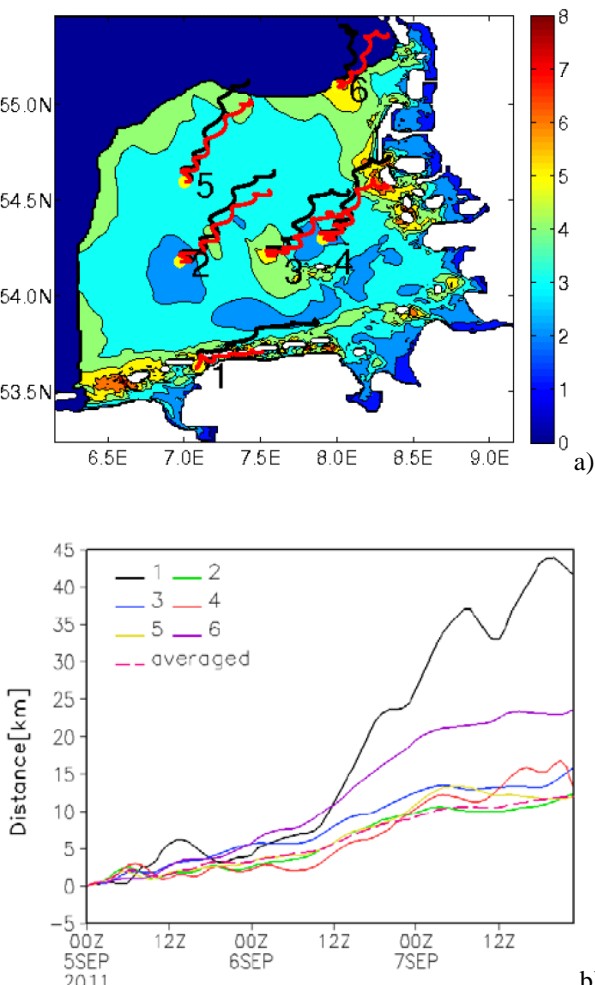

a)

b)

Figure 10. (a) The displacement of Lagrangian particles. Black lines give the results from the model free run, red trajectories visualize the results in the data assimilation run. Colour coding gives the mean distance in km between position of Lagrangein particles in the analysis and free run for September 2011 after 24h of integration. (b) Distance of drifting particles in the assimilation run and the free run.

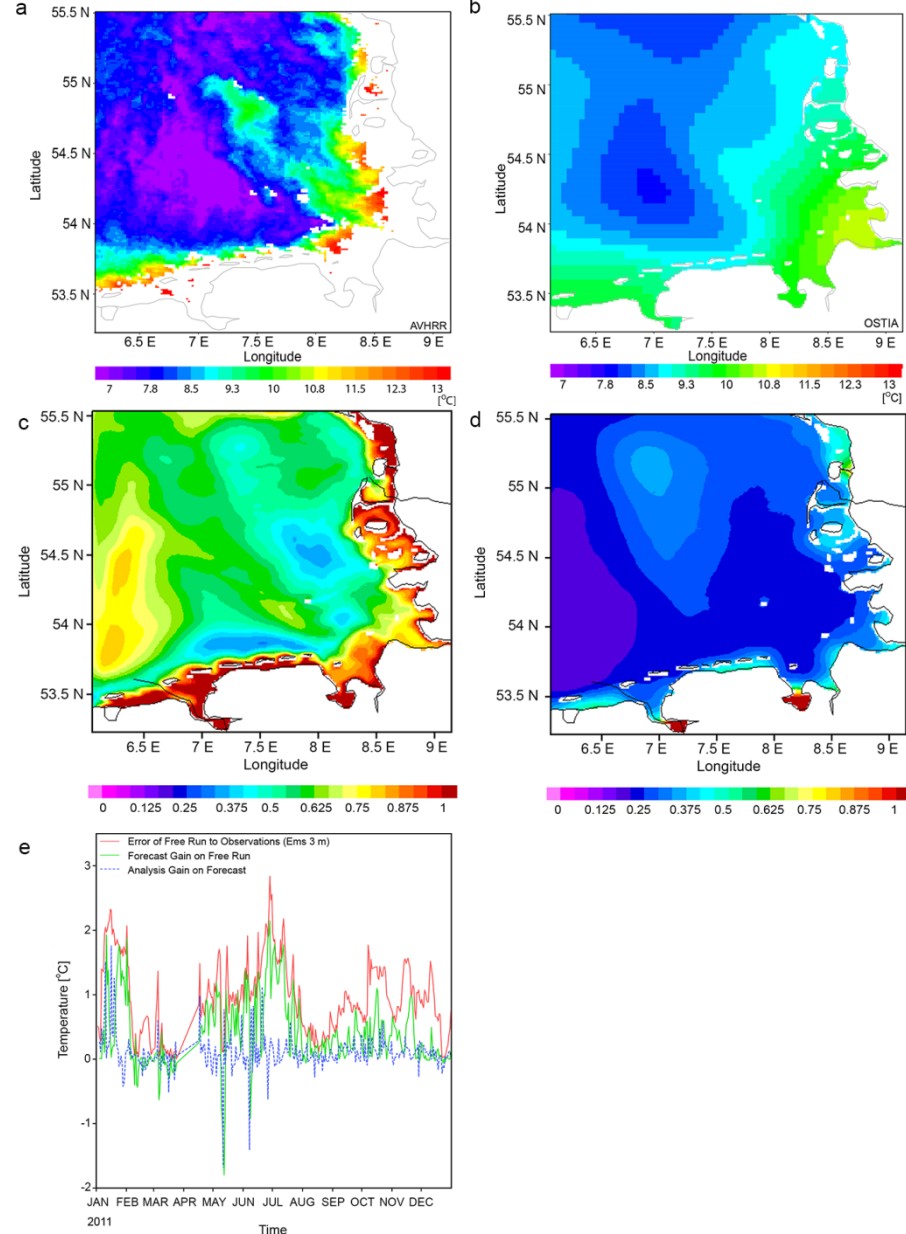

Figure 11. (a)  SST from the L3 multi-sensor super-collated SST product (CMEMS 2015) and OSTIA data (b) for 23.04.2011. (c) RMSE between OSTIA SST and the numerical simulations from the free run of the German Bight model (see Annex 6.2) for 2011. (d) As (c), but with assimilation of OSTIA SST data. (e) Validation statistics of data assimilation

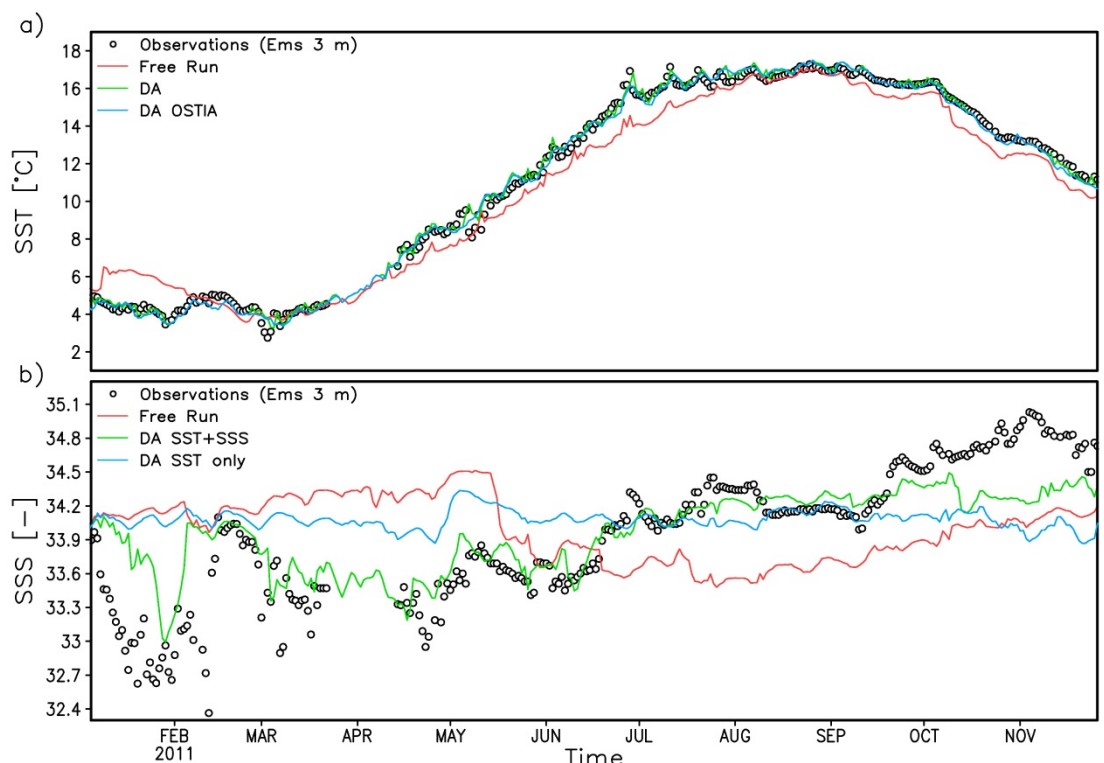

Figure 12. Validation of the simulated with the German Bight model SST (a) and SSS (b) against the MARNET observations during 2011 for the Ems data station (see Fig. 1 for its position). Black circles are MARNET observations, the red line is the free run, the blue line corresponds to the case when only SST is assimilated, the green line shows the results when the assimilation of SST and SSS from Ferry box is added. Temporal resolution of the data is hourly and it is smoothed with a 24 hours running mean in order to filter out the daily cycle. The different experiments and data are shown in the legend.

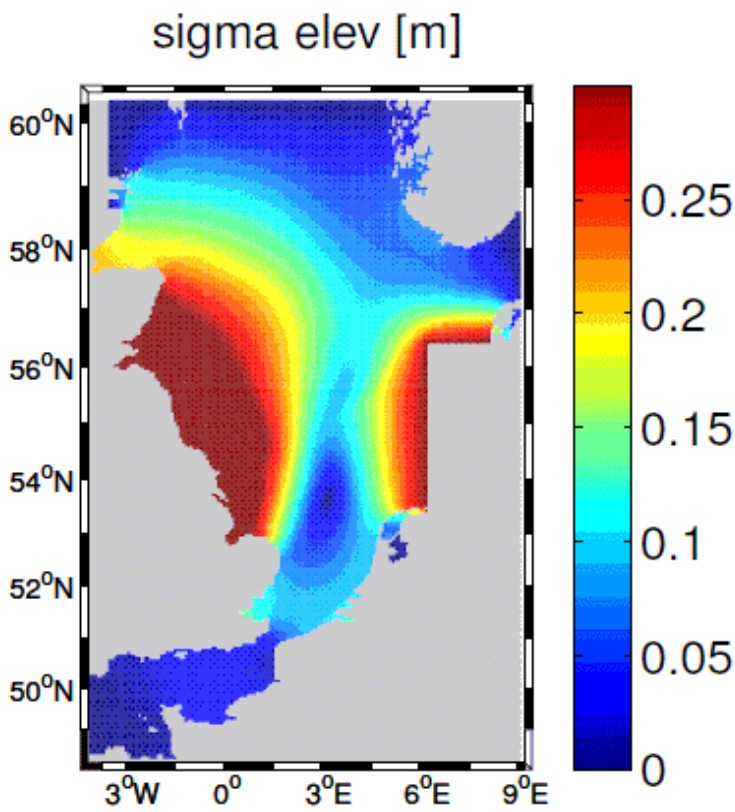

Figure 13. Impact of coastal boundary condition of a nested model on the North Sea dynamics. The figure originates from Schulz-Stellenfleth and Stanev (2016) and shows the standard deviation of elevation resulting from fully correlated perturbations with stdv 0.3 m along the German Bight border.

