# Peer review of "Ocean Forecasting for the German Bight: From Regional to Coastal Scales"

_Ocean Science, 2016_

## Referee Comment (RC1) · Anonymous Referee #1 · 1 Jun 2016

General comments ============

This paper provides a review of research work at HZG in the field of coastal oceanography in the German Bight, as part of a special collection of papers. This manuscript highlights and discusses a number of research issues relevant to others currently working in this field, and covers a lot of research angles aimed at improve model predictions. Contrasts between coastal and open ocean approaches are provided clearly, which is welcome. The paper is generally well written, and I would recommend it for publication following some minor updates.

The manuscript could be generally improved by making linkages and dependencies between sections a little clearer, so that the whole paper hangs together a little better as a coherent discussion. A few specific illustrations and suggestions are provided in

the specific comments below which may help to address this.

The authors set out two main goals: to showcase methodologies integrating observations and models in coastal areas, and to provide an analysis of the synergy of coastal and larger-scale forecasting systems. These aims are largely met.

Specific comments ============

Title – given the quite specific regional focus of research work covered in this paper on the German Bight, I would ask the authors to consider a more specific title for this paper, e.g. "Ocean Forecasting for the German Bight: From Regional to Coastal Scales" as a more descriptive title.

Section 1, please clarify further how your paper adds beyond other review papers (e.g. Kourafalou et al 2015a,b)

Section 2 – this whole section would benefit from clearer sign-posting of what is covered within each section, for example via a short introduction on p4, and clearer titles for sub-sections. At present it reads as a slightly ad-hoc list of different approaches and evaluations of improving model skill. It would aid the reader to understand how 2.2 links to 2.3? Should Section 2.3 better sit in its own section on model nesting?

Section 2.1 – on discussing "resolution capacity compliant with the dominant spatial scales", it would aid the reader to add another line of detail relating to how that choice can be sensibly made (e.g. consideration of the relevant Rossby radius of deformation? What are the relevant length scales for the German Bight?). Is it also possible to discuss sensible choices for vertical resolution in these domains?

Section 2.2.3 and 2.2.4 – to aid discussion of the different model configurations compared here, it would be helpful to provide a summary table (summarising the annex material) which highlights the key differences between systems shown. E.g. are all systems operating with the same horizontal and vertical resolution, Baltic model, atmospheric forcing, freshwater fluxes, etc?

Section 2.2.4 – please clarify whether differences in M4 tides are purely a function of different model resolutions, or are there other factors?

Section 2 – freshwater fluxes: there is no mention in this section of the importance of accurate freshwater fluxes for prediction in the coastal ocean (or indeed whether this is an issue in the German Bight). It could be helpful to the reader to provide a brief discussion on this, particularly in light of Section 4.3.

Section 3.2 – HF radar. Please provide summary statistics of the value of HF radar within the COSYNA system. It is difficult for the reader to understand the value of these data from the discussion alone as it stands (see also comment re. Figure 4). This is more complete in the discussion of SST assimilation.

Figure 4 – Can the authors provide any longer-term analysis of HF radar data vs model analyses and free run? E.g. long-term statistics (as provided in Figure 5 for example).

Section 3.3 – please comment on the errors in OSTIA in the coastal zone, given its dependence on satellite products for which errors are increased here. To what extent would the authors expect assimilation of OSTIA to provide information on the detailed structures in the coastal zone? This seems particularly relevant to the discussion relative to DA_BLEND results.

Figure 5 – while Figure 4 refers to a snapshot comparison (see above comment), it would be valuable to compare snapshots, or some assessment on sub-annual timescale of differences between OSTIA and the numerical model, to compare how well captured the near-coastal variability might be between OSTIA and the model.

Section 4.3 – it would be helpful to better link this section in to the preceding discussion. The key question, is how does the estuary-specific configuration interface with the larger-scale German Bight models, if at all, and what are the challenges to address in nesting right across scales from North Sea to estuary scale? Is this a 'solved' issue? It is currently difficult to understand how the Ems Estaury model fits relative to other

tools available to provide services.

Section 4.6 – please provide some context for the quantitative differences discussed. E.g. is a 40cm difference important for end-users and responding to natural hazards? How do underestimation of 30cm relative to gauge compare with long-term statistics for sea level predictions in this region – is this specific to extreme events or typical?

Technical corrections =============

P2, Para including line 20: "….similar devastations never happened again.". Please consider addressing the language in this sentence to something like ….."similar devastation has not occurred since" – there are of course a number of reasons for this (e.g. have similar magnitude storms hit the region since?).

P2 – check citations Kourafalou et al 2014 or 2015?

P3, line 25 – please check language concerning "data problematics", suggest rephrasing this point.

P9, line 5 – please clarify status of "in preparation" paper ahead of publication

P15, line 10 – please check if "no-seamless" is a typing error, or if a clearer phrase can be used P15, para beginning line 20 – typo "Lagrangean"?

---

## Referee Comment (RC2) · Anonymous Referee #2 · 8 Jun 2016

The paper "Ocean Forecasting: From Regional to Coastal Scales" describes various ocean modeling applications using data from the COSYNA observing system. The main purpose of the paper is "to showcase methodologies integrating observations and models in coastal areas", with a secondary objective being to present an "analysis of the synergy of coastal and larger-scale forecasting systems".

My main objection to the present ms is that it lacks structure and does not present anything really new. Cursory examples are presented, with little to no supporting long-term statistics to back up the various conclusions. As a review paper it is too focused on the shallow water dynamics in one particular region and its very general title is not justified.

In my opinion, the interesting parts of the paper are the discussions about tides and

storm surge, which is certainly important in the region of interest. I recommend that the ms in its present form is rejected and that the authors instead resubmit a more focused study on the shallow water dynamics in this region, with more emphasis on verification and less emphasis on specific examples.

Specific comments:

- The presentation is confusing and the text is not properly structured. Again, restricting focus to one specific dynamical problem would help increase the clarity of the presentation. Incosistent use of abbreviations adds to the confusion (e.g. "SAR" vs "search and rescue").

- Central information about the various modeling systems is only given in the appendix. The level of detail is unsatisfactory and the ms cannot stand on its own in its present form. A proper model comparison will require a more elaborate discussion about their differences, for instance the impact of using hourly vs six hourly atmospheric forcing. It is also difficult to keep track of which model is used for what purpose as the authors jump back and forth between them in the examples.

- Errors of representativeness, which becomes an important issue when downscaling data assimilative models merits a discussion, but is not mentioned here.

- References are missing in several places, e.g. pages/lines 2/15, 3/24, 7/12; there are several errors in the citations (e.g. 5/24, 10/28); and reference to unpublished material makes no sense (9/4).

- The HF radar assimilation technique based on the method of Stanev et al (2015) may be well justified for use in this region, but might be less useful in regions where baroclinicity and/or the influence of complex topography dominates. It would be good to see an assessment of the impact of HF radar DA on storm surge predictions instead of the (very short) discussion about search-and-rescue support. Several published papers deal with the impact of HF radar data assimilation on current predictions, e.g.

[Figure]

Barth et al (2008, JGR, using ADCP for verification), Yaremchuck et al (2016, DSR II, using drifters), Sperrevik et al (2015, OS, using both drifters and ADCP), so that the cursory example presented here does not really provide anything new.

- The apparently small impact of in-situ data (ferrybox) vs the OSTIA product indicates that the DA system is not working optimally. I would expect in-situ data to be rather more valuable, but again, very little in the way of statistics is presented, e.g. innovations vs analysis increments and their temporal and spatial distributions. Mention could also be made about rapid update cycles, which is used successfully by e.g. the KNMI in their regional NWP system to maximise the use of observations in small model domains (deHaan, 2013, QJRMS).

- The "two-way nesting" method described in Sec. 2.3 differs from the full online nesting implemented in e.g. ROMS and AGRIF. I assume the nudging based method presented here will in practice work as a low pass filter when information is exchanged between parent and child grids, and I would like to see what the impact is on fast time scales such as tidal wave propagation.

---

## Referee Comment (RC3) · Anonymous Referee #3 · 4 Jul 2016

The main purpose of the manuscript entitled "Ocean Forecasting: From Regional to Coastal Scales" is to present various modelling applications mainly by HZG in order to elucidate different interesting scientific aspects of ocean forecasting at the coastal scale. Although its style refers to a review paper about ocean forecasting on the coastal scale it mainly (if not only) refers to the COSYNA observing system and various applications over the German Bight. The manuscript contains a lot of information (from Data assimilation to tides, wave – current interactions estuarine and search & rescue applications) which in most situations is not well structured/organized and sometimes becomes quite confusing for the reader. Moreover no mention at all for the effect of atmospheric forcing in coastal forecasting is given. I think a whole subsection should be devoted to this important for coastal applications aspect. Along this line air-sea interaction and issues related to wave current interactions (for example the momentum

and energy surface boundary condition) should be discussed in more detail. I think that the authors should concentrate on mostly 2-3 topics (for example data assimilation of HF Radar or satellite/in-situ SST data on the coastal scale and wave –current interactions) instead of overwhelming the reader with excessive material which is not complete (for example in section 4.6 where the important topic of wave – current interactions is involved/discussed the reader is just referred the paper by Staneva et al., 2015 for the scientific approach & discussion) and cannot be easily digested. In this sense, I propose a major revision of the present manuscript with drastic restructuring and focusing on a much more limited list of topics related to coastal forecasting.

Specific comments:

-The title of the manuscript should contain the toponym "German Bight". I agree with the new title proposed by the anonymous referee #1

-Section 3.1: The approach proposed to overcome the situation where the assimilation degrades the model results due to hf perturbations, is never presented explicitly in this paper.

-Section 3.3: what do we see in fig. 5b? The analysis RMSE? If yes I would prefer to judge the performance of the assimilation system by checking the forecast RMSE. In any case a more in depth analysis of the results is needed in order to understand the impact of OSTIA and in-situ observations.

-Section 4.4 can be omitted. I do not understand its role in this manuscript.

-Section 4.6: more in depth presentation and analysis of the results is needed.

---

## Author Comment (AC1) · 20 Aug 2016

Answers on "Ocean Forecasting: From Regional to Coastal Scales" by Emil V. Stanev et al.

Anonymous Referee #1

We are grateful to reviewer for the appreciation of our work and his constructive comments, which we answer point-by-point. The comments, which are not included in our answers are of technical character, and are addressed in the revised manuscript as the reviewer suggested.

The manuscript could be generally improved by making linkages and dependencies between sections a little clearer, so that the whole paper hangs together a little better

as a coherent discussion.

Authors: In the revised manuscript we include linkages and dependencies between sections, as suggested.

Title – given the quite specific regional focus of research work covered in this paper on the German Bight, I would ask the authors to consider a more specific title for this paper, e.g. "Ocean Forecasting for the German Bight: From Regional to Coastal Scales" as a more descriptive title.

Authors: Thank you for this suggestion. We accept it.

Section 1, please clarify further how your paper adds beyond other review papers (e.g. Kourafalou et al 2015a,b)

Authors: At the end of the revised Introduction we include a paragraph clarifying this. The major point is that we consider only one specific region (the North Sea and Baltic Sea) and focus on the transitions between coastal and regional scale as seen in the analyses for two different transition areas: German Bight and straits connecting the Baltic and North Seas.

Section 2 – this whole section would benefit from clearer sign-posting of what is covered within each section, for example via a short introduction on p4, and clearer titles for sub-sections. At present it reads as a slightly ad-hoc list of different approaches and evaluations of improving model skill.

Authors: The referee is right and we are thankful for this suggestion. We introduce in the beginning of each major section a description of research issues which are addressed and explain the rationale, which justifies keeping the individual sub-sections together. In the revised manuscript substantial restructuring has been done. (1) We explain better the links between individual parts of paper. (2) We removed sub-sections 4.1 and 4.4. (3) We moved sub-section 4.5 into section 3 and explained why. (4) We reordered the remaining part of section 4 starting with coupled wave-circulation

modelling. (5) We re-structured section 2.3. (6) We changed the titles of some sub-sections.

It would aid the reader to understand how 2.2 links to 2.3? Should Section 2.3 better sit in its own section on model nesting?

Authors: In the revised manuscript we explain the idea behind keeping Section 2.3 (which is on modelling) in the modelling part. In the revised paper (end of introduction), the structure of paper is also explained.

Section 2.1 – on discussing "resolution capacity compliant with the dominant spatial scales", it would aid the reader to add another line of detail relating to how that choice can be sensibly made (e.g. consideration of the relevant Rossby radius of deformation? What are the relevant length scales for the German Bight?). Is it also possible to discuss sensible choices for vertical resolution in these domains?

Authors: In the revised manuscript we address this comment and provide the missing information and references.

Section 2.2.3 and 2.2.4 – to aid discussion of the different model configurations compared here, it would be helpful to provide a summary table (summarising the annex material) which highlights the key differences between systems shown. E.g. are all systems operating with the same horizontal and vertical resolution, Baltic model, atmospheric forcing, freshwater fluxes, etc?

Authors: In the revised manuscript we provide in section 2.2.2 a new table summarizing the annex material and add a synthesis of model characteristics presented in the Annex.

Section 2.2.4 – please clarify whether differences in M4 tides are purely a function of different model resolutions, or are there other factors?

Authors: Good point, which is very important for the substance of the paper.

1. When describing the specificity of coastal ocean modelling in section 2.1 we add a paragraph on this explaining that it is not only the resolution in the models, which maters when moving from regional to coastal scales, but also the details in bathymetry, such as the coast-line and bottom roughness, which could change in time. Addressing specific processes and their role in the coastal ocean is basic in order to understand whether by just changing the resolution we could solve the major problems with the transition between the coastal and regional scales. On the road of this transition specific processes, which are sometimes neglected in global and regional forecasting, start to dominate .One second example demonstrating the role of surface waves is presented in section 4.1 (old section 4.6).

2. We address this issue in the revised manuscript in more specific terms in Section 2.2.4 (M4 tides) where we mention that according to the analysis of data and modelling results of Jacob et al. (2016) the morphodynamics could result in a substantial migration of bottom channels in the Wadden Sea. This can be considered as a change in the macro-scale bottom roughness, which triggers non-local responses. In the context of major issue addressed in this section, one needs to mention that response to bottom changes identified by Jacob et al. (2016) is quite strong for the M4 tide, giving further motivation to deepen the understanding of properties of shallow-water tides.

Section 2 – freshwater fluxes: there is no mention in this section of the importance of accurate freshwater fluxes for prediction in the coastal ocean (or indeed whether this is an issue in the German Bight). It could be helpful to the reader to provide a brief discussion on this, particularly in light of Section 4.3.

Authors:

1. After "In addition, more flexible coupling is needed between regional and coastal models, including estuarine models."(Section 2.1 of the first submission) we mention the issue about the river runoff.

2. In Section 2.2 when describing the general characteristics of models we focus also

on possible problems with the representation of river run, referring to section 4.2 (old section 4.3) where we discuss this issue in more detail. The importance of this for the North Sea and Baltic Sea is also mentioned there.

Section 3.2 – HF radar. Please provide summary statistics of the value of HF radar within the COSYNA system. It is difficult for the reader to understand the value of these data from the discussion alone as it stands (see also comment re. Figure 4). This is more complete in the discussion of SST assimilation.

Authors: In the revised manuscript we provide summary statistics (new table) and text summarizing the statistics.

Figure 4 – Can the authors provide any longer-term analysis of HF radar data vs model analyses and free run? E.g. long-term statistics (as provided in Figure 5 for example).

Authors: In the revised manuscript we provide some new summary statistics (new table).

Section 3.3 – please comment on the errors in OSTIA in the coastal zone, given its dependence on satellite products for which errors are increased here. To what extent would the authors expect assimilation of OSTIA to provide information on the detailed structures in the coastal zone? This seems particularly relevant to the discussion relative to DA_BLEND results.

Authors:

1. We refer in the revised manuscript to our previous publication where the errors in OSTIA data are presented.

2. In the revised manuscript we address issue about benefit of assimilating OSTIA data in the coastal area admitting that the major impact of OSTIA data assimilation is in the improvement of the large-scale temporal and spatial characteristics.

Figure 5 – while Figure 4 refers to a snapshot comparison (see above comment),
it would be valuable to compare snapshots, or some assessment on sub-annual timescale of differences between OSTIA and the numerical model, to compare how well captured the near-coastal variability might be between OSTIA and the model.

Authors: We make clear in the revised manuscript that neither OSTIA data nor the numerical model with a resolution of 1 km can well resolve near-coastal variability. Just to visualize the problems with resolving small-scale features in OSTIA data we add to Fig. 5 new frames to illustrate OSTIA data and fine-resolution observations and free model run for the same time. The problems with resolving small scales are used to bridge the results in this section with the ones where discuss simulations of near-coastal zone and estuaries.

Section 4.3 – it would be helpful to better link this section in to the preceding discussion. The key question, is how does the estuary-specific configuration interface with the larger-scale German Bight models, if at all, and what are the challenges to address in nesting right across scales from North Sea to estuary scale? Is this a 'solved' issue? It is currently difficult to understand how the Ems Estaury model fits relative to other tools available to provide services.

Authors:

1. The title of this section has been changed.

2. The first part of this section was written new, addressing the comment of the referee.

3. The focus of the presentation of results has been also changed accordingly, to link this section in to the preceding discussion about the consistence between the estuary-specific modelling and the larger-scale German Bight model.

4. More weight in the revised section is given to the transformation of fresh water in the estuary and beyond.

Section 4.6 – please provide some context for the quantitative differences discussed. E.g. is a 40cm difference important for end-users and responding to natural hazards?

How do underestimation of 30cm relative to gauge compare with long-term statistics for sea level predictions in this region – is this specific to extreme events or typical?

Authors: In the revised manuscript we address this issue with respect of the uncertainties of storm surge predictions and the quantification of associated coastal hazards. We stress that the results of our experiments showed that the wave-dependent approach yields to ∼30% larger surge for the period of "Xavier".

P2, Para including line 20: ". . .similar devastations never happened again.". Please consider addressing the language in this sentence to something like . . .."similar devastation has not occurred since" – there are of course a number of reasons for this (e.g. have similar magnitude storms hit the region since?).

Authors: We rephrased this sentence.

P3, line 25 – please check language concerning "data problematics", suggest rephrasing this point.

Authors: We rephrased this sentence.

P9, line 5 – please clarify status of "in preparation" paper ahead of publication

Authors: We removed this reference.

---

## Author Comment (AC2) · 20 Aug 2016

Answers on "Ocean Forecasting: From Regional to Coastal Scales" by Emil V. Stanev et al.

Anonymous Referee #2

We are grateful to reviewer for the appreciation of some parts of our work and his constructive comments, which we answer point-by-point. Some comments of technical character, which are not included in our answer, are addressed in the revised manuscript as the reviewer suggested.

My main objection to the present ms is that it lacks structure and does not present anything really new. Cursory examples are presented, with little to no supporting longterm

statistics to back up the various conclusions. As a review paper it is too focused on the shallow water dynamics in one particular region and its very general title is not justified.

Authors:

1. In the revised manuscript we improve the structure of the paper with developing the logical links between its sections, as this was proposed also by the first referee.

2. We admit that an impression (missing novelty) could have occurred because we did not enough stress on what is the new development. In the revised paper we made clear what the novelties are.

3. In the revised manuscript we provide new statistics in form of tables and graphics to support our conclusions.

4. We made clear in the revised manuscript that the paper is about short-term predictions, not long-term ones.

5. This paper has been submitted to the COSYNA special issue of Ocean Science. Therefore we focus on the areas where most of activities of COSYNA take place that is in "one particular region". We want also to mention that it is not possible in one paper to address in sufficient detail many different coastal areas. Our choice was one area, but several different aspects.

6. Following the comments of referees we changed the title.

I recommend . . . that the authors instead resubmit a more focused study on the shallow water dynamics in this region, with more emphasis on verification and less emphasis on specific examples.

Authors:

1. The first submission was exclusively on the shallow water dynamics.

2. We provide in the revised manuscript more verification material, numbers for statis-
tics, tables, etc. and respective explanations.

3. The number of examples considered has been reduced.

4. The presentation is confusing and the text is not properly structured. Again, restricting focus to one specific dynamical problem would help increase the clarity of the presentation.

Incosistent use of abbreviations adds to the confusion (e.g. "SAR" vs "search and rescue").

Authors:

1. In the revised manuscript we develop the logical links between individual parts of the paper.

2. We formulate for each sub-section more clearly what is the specific dynamical problem addressed.

3. The presentation has been restructured (sub-sections omitted, other sub-sections displaced, some sub-sections are restructured, in some others more weight has been given to issues suggested by referees).

4. We avoid using misleading abbreviations.

- Central information about the various modeling systems is only given in the appendix. The level of detail is unsatisfactory and the ms cannot stand on its own in its present form. A proper model comparison will require a more elaborate discussion about their differences, for instance the impact of using hourly vs six hourly atmospheric forcing.

Authors:

1.As suggested also by the referee #1, we present in the revised manuscript a table of models used in the paper and their most important details.

2. We increase (wherever necessary) the level of details given for the individual models
in order to balance the deepness of presentation of all models almost the same.

3. As we stated in the first submission, model inter-comparison is not the aim of this study. Explanation about this strategy has been presented in the first submission (see section 2.2.2 where the rationale is presented). We want to remind that outputs of some operational models have been used and it is not possible to change the way how atmospheric forcing has been used.

4. Our strategy was to use forcing data with as fine as possible resolution in time.

It is also difficult to keep track of which model is used for what purpose as the authors jump back and forth between them in the examples.

Authors: We checked carefully all sections and wherever this has not been explained clearly, we provide the necessary information.

- Errors of representativeness, which becomes an important issue when downscaling data assimilative models merits a discussion, but is not mentioned here.

Authors: In the revised manuscript we provide new skill estimates (in particular as far as data assimilation has been concerned ) and refer to previous studies on this.

- References are missing in several places, e.g. pages/lines 2/15, 3/24, 7/12; there are several errors in the citations (e.g. 5/24, 10/28); and reference to unpublished material makes no sense (9/4).

Authors: We are thankful for this comment and we did all proposed corrections in the revised manuscript.

- The HF radar assimilation technique based on the method of Stanev et al (2015) may be well justified for use in this region, but might be less useful in regions where baroclinicity and/or the influence of complex topography dominates.

Authors: We focus on the German Bight in this paper. Addressing other regions would increase the diversity of addressed issues, which we, following referees' comments,
want to avoid. Following this comment we mention in the text that the demonstration of skill is valid for the specific area. Further applications of proposed method to different regions (e. g., regions dominated by pronounced baroclinicity) need additional analysis. As said above in the limited space of this paper we cannot address many different areas, also because availability of HF radar data is also a problem.

It would be good to see an assessment of the impact of HF radar DA on storm surge predictions instead of the (very short) discussion about search-and-rescue support.

Authors: In the revised paper we address the capability of HF radar data to detect changes in surface currents during storm surge periods. In this way we also increase the inter-connectivity between different sections.

Several published papers deal with the impact of HF radar data assimilation on current predictions, e.g. Barth et al (2008, JGR, using ADCP for verification), Yaremchuck et al (2016, DSR II, using drifters), Sperrevik et al (2015, OS, using both drifters and ADCP), so that the cursory example presented here does not really provide anything new.

Authors:

1. We stressed in the first submission that the novelty we address is in the forecasting at intra-tidal time scales. Most of the past studies use de-tided HFR data. In the revised manuscript we emphasize on this novelty (as this was one of the suggestions in the general comments).

2. In section 3.2 we include a short presentation of earlier works in this field and cite the proposed ones.

- The apparently small impact of in-situ data (ferrybox) vs the OSTIA product indicates that the DA system is not working optimally. I would expect in-situ data to be rather more valuable, but again, very little in the way of statistics is presented, e.g. innovations vs analysis increments and their temporal and spatial distributions. Mention could also be made about rapid update cycles, which is used successfully by e.g. the KNMI in their

regional NWP system to maximise the use of observations in small model domains (deHaan, 2013, QJRMS).

Authors:

1. We explain in the revised manuscript the reasons of "small impact of in-situ data" providing more statistics.

2. We show a comparison between free run, OSTIA and fine-resolution temperature data.

3. Additionally we refer to earlier publications (Grayek et al., 2011; Stanev et al., 2011) where more details are given about the systems' performance and skill estimates.

4. We refer in the revised manuscript to the publication mentioned by the referee.

- The "two-way nesting" method described in Sec. 2.3 differs from the full online nesting implemented in e.g. ROMS and AGRIF. I assume the nudging based method presented here will in practice work as a low pass filter when information is exchanged between parent and child grids, and I would like to see what the impact is on fast time scales such as tidal wave propagation.

Authors: Answering this question, we provide in the revised manuscript more details about temporal variability (tidal analysis and variability of salinity in the transition area), which demonstrates that the proposed method does not the impact negatively dominant dynamics in the transition area.

―――――――――――――――

---

## Author Comment (AC3) · 20 Aug 2016

Answers on "Ocean Forecasting: From Regional to Coastal Scales" by Emil V. Stanev et al.

Anonymous Referee #3

We are grateful to reviewer for the appreciation of some parts of our work and his constructive comments, which we answer point-by-point.

The manuscript contains a lot of information (from Data assimilation to tides, wave – current interactions estuarine and search & rescue applications) which in most situations is not well structured/organized and sometimes becomes quite confusing for the reader.

Authors: In the revised manuscript substantial restructuring has been done. We explain better the links between individual parts of paper and, removed sub-sections 4.1 and 4.4, moved sub-section 4.5 into section 3 and explained why; reordered the remaining part of section 4 starting with coupled wave-circulation modelling, changed the titles of some sub-sections.

Moreover no mention at all for the effect of atmospheric forcing in coastal forecasting is given. I think a whole subsection should be devoted to this important for coastal applications aspect. Along this line air-sea interaction and issues related to wave current interactions (for example the momentum and energy surface boundary condition) should be discussed in more detail.

Authors: There are some reasons not to devote one separate section to the atmospheric forcing. In the revised manuscript, we rather add some references to studies on this subject carried out in the past (Backhaus, 1989; Skogen et al., 2011, Dangendor et al., 2014). Another argument for this follows the suggestions to keep this manuscript more focused. The third one was to demonstrate novel developments, and we consider the issue about shallow-water tides as one such issue. In the revised paper we integrated the issue of atmospheric forcing with the novel development of coupled wave-current modelling. Additionally a more detailed presentation on the coupling method is given.

I think that the authors should concentrate on mostly 2-3 topics (for example data assimilation of HF Radar or satellite/in-situ SST data on the coastal scale and wave – current interactions) instead of overwhelming the reader with excessive material which is not complete (for example in section 4.6 where the important topic of wave – current interactions is involved/discussed the reader is just referred the paper by Staneva et al., 2015 for the scientific approach & discussion) and cannot be easily digested.

Authors:

1. In the revised manuscript we provide more coherent and complete presentation of

OSD
the material.

2. Section 4.6 has been reshaped in lines with what referee suggested.

In this sense, I propose a major revision of the present manuscript with drastic restructuring and focusing on a much more limited list of topics related to coastal forecasting.

Authors: As said above, we restructured the paper, removed parts (which are not so closely related to the coastal forecasting) and re-focused, as suggested by referee. To our understanding all aspects considered in the resubmitted paper are linked to coastal forecasting, and we hope that we expressed this in a more convincing way now.

-The title of the manuscript should contain the toponym "German Bight". I agree with the new title proposed by the anonymous referee #1

Authors: The title has been changed as suggested by referee#1: "Ocean Forecasting for the German Bight: From Regional to Coastal Scales".

-Section 3.1: The approach proposed to overcome the situation where the assimilation degrades the model results due to hf perturbations, is never presented explicitly in this paper.

Authors: We present this issue in the revised manuscript.

-Section 3.3: what do we see in fig. 5b? The analysis RMSE? If yes I would prefer to judge the performance of the assimilation system by checking the forecast RMSE.

Authors: Fig 5 has been changed and more extensive analysis presented.

In any case a more in depth analysis of the results is needed in order to understand the impact of OSTIA and in-situ observations.

Authors: We extend this analysis in the revised manuscript commenting also on the high resolution radiometer data.

-Section 4.4 can be omitted. I do not understand its role in this manuscript.

OSD
Authors: We omitted this section.

-Section 4.6: more in depth presentation and analysis of the results is needed.

Authors: We substantially revised this part presenting in more detail the coupled model and the analysis of simulations.

---

## Author Response (AR1)

**Answers of reviewers' comments on "Ocean Forecasting: From Regional to Coastal Scales" by Emil V. Stanev et al.**

**Anonymous Referee #1**

We are grateful to reviewer#1 for the appreciation of our work and his constructive comments (in italic below), which we answer point-by-point. The comments, which are not included in our answers are of technical character, and are addressed in the revised manuscript as the reviewer suggested.

The manuscript could be generally improved by making linkages and dependencies between sections a little clearer, so that the whole paper hangs together a little better as a coherent discussion.

Authors: In the revised manuscript we include linkages and dependencies between sections, as suggested.

Title – given the quite specific regional focus of research work covered in this paper on the German Bight, I would ask the authors to consider a more specific title for this paper, e.g. "Ocean Forecasting for the German Bight: From Regional to Coastal Scales" as a more descriptive title.

Authors: Thank you for this suggestion. We accept it.

Section 1, please clarify further how your paper adds beyond other review papers (e.g. Kourafalou et al 2015a,b)

Authors: At the end of Introduction we include two paragraphs clarifying this (please see the last two paragraphs in the revised Introduction). The novelty of this study has been stressed many times when discussing new results.

Section 2 – this whole section would benefit from clearer sign-posting of what is covered within each section, for example via a short introduction on p4, and clearer titles for sub-sections. At present it reads as a slightly ad-hoc list of different approaches and evaluations of improving model skill.

Authors: The referee is right and we are thankful for this suggestion. We introduce in the beginning of each major section a description of research issues which are addressed and explain the rationale, which justifies keeping the individual sub-sections together. In the revised manuscript substantial restructuring has been done. We explain better the links between individual parts of paper and, removed sub-sections 4.1 and 4.4, moved sub-section 4.5 into the new section 3. Old section 4.6 was moved into the modelling part (new section 2). We also changed the titles of some sub-sections.

It would aid the reader to understand how 2.2 links to 2.3? Should Section 2.3 better sit in its own section on model nesting?

Authors: In the revised manuscript we explain the idea behind keeping Section 2.3 (which is on modelling) in the modelling part. In the revised paper (end of introduction), the structure of paper is also explained.

Section 2.1 – on discussing "resolution capacity compliant with the dominant spatial scales", it would aid the reader to add another line of detail relating to how that choice can be sensibly made (e.g. consideration of the relevant Rossby radius of deformation? What are the relevant length scales for the German Bight?). Is it also possible to discuss sensible choices for vertical resolution in these domains?

Authors: In the revised manuscript we address this comment and provid the missing information and references (please, see the second paragraph of section 2.1).

Section 2.2.3 and 2.2.4 – to aid discussion of the different model configurations compared here, it would be helpful to provide a summary table (summarising the annex material) which highlights the key differences between systems shown. E.g. are all systems operating with the same horizontal and vertical resolution, Baltic model, atmospheric forcing, freshwater fluxes, etc?

Authors: In the revised manuscript we provide in section 2.2.2 a new table summarizing the Annex material and add a synthesis of model characteristics presented in the Annex.

Section 2.2.4 – please clarify whether differences in M4 tides are purely a function of different model resolutions, or are there other factors?

Authors: Good point, which is very important for the substance of the paper.

When describing the specificity of coastal ocean modelling in section 2.1 we add a paragraph on this explaining that (citation) It is not only the spatial resolution that maters when moving from the regional to coastal scale but also the details of bathymetry, such as the coastline and bottom roughness, the latter of which can also change in time. Addressing specific processes and their role in the coastal ocean is essential to understand whether we could solve the major problems with the transition between the regional and coastal scales by only changing the resolution. Specific processes, e.g., shallow-water tides, which are sometimes neglected in global and regional forecasting, dominate coastal ocean dynamics. An additional example that demonstrates the role of surface waves in the coastal zone is presented in section 2.4.

We address this issue in the revised manuscript in in the last paragraph of section 2.2.4 (M4 tides) where we mention role of morphodynamics and the sensitivity of the M4 tide to it.

Section 2 – freshwater fluxes: there is no mention in this section of the importance of accurate freshwater fluxes for prediction in the coastal ocean (or indeed whether this is an issue in the German Bight). It could be helpful to the reader to provide a brief discussion on this, particularly in light of Section 4.3.

**Authors:**

- After "In addition, more flexible coupling is needed between regional and coastal models, including estuarine models." (Section 2.1 of the first submission) we mention in the revised manuscript he issue about the river runoff, and where this is addressed in the revised manuscript. A large part of the new section 2.5 addresses this issue.
- The last sentence in the introduction to section 2 says "Section 2.5, which addresses estuarine seamless modelling and quantifies the pattern of water mass transformation between rivers and open ocean, can be considered as a step towards linking estuarine and regional ocean modelling."
- The importance of river runoff for the North Sea and Baltic Sea is mentioned in section 2.1

Section 3.2 – HF radar. Please provide summary statistics of the value of HF radar within the COSYNA system. It is difficult for the reader to understand the value of these data from the discussion alone as it stands (see also comment re. Figure 4). This is more complete in the discussion of SST assimilation.

Authors: In the revised manuscript we provide summary statistics (new Table 1 and Table 2) and text summarizing the statistics.

*Figure 4 – Can the authors provide any longer-term analysis of HF radar data vs model analyses and free run? E.g. long-term statistics (as provided in Figure 5 for example).*

Authors: In the revised manuscript we provide new summary statistics Table 2, showing innovations and analysis residuals. Furthermore, the skill of data assimilation with regard to independent ADCP data is demonstrated in Table 3.

Section 3.3 – please comment on the errors in OSTIA in the coastal zone, given its dependence on satellite products for which errors are increased here. To what extent would the authors expect assimilation of OSTIA to provide information on the detailed structures in the coastal zone? This seems particularly relevant to the discussion relative to DA\_BLEND results.

Authors:

- We refer in the revised manuscript to our previous publication where the errors in OSTIA data are presented.
- In the revised manuscript we address the issue about benefit of assimilating OSTIA data in the coastal area admit that the major impact of OSTIA data assimilation is in the improvement of the large-scale temporal and spatial characteristics.

Figure 5 – while Figure 4 refers to a snapshot comparison (see above comment), it would be valuable to compare snapshots, or some assessment on sub-annual timescale of differences between OSTIA and the numerical model, to compare how well captured the near-coastal variability might be between OSTIA and the model.

Authors: We make clear in the revised manuscript that, neither OSTIA data, nor the numerical model with a resolution of 1 km can well resolve near-coastal variability. Just to visualize the problems with resolving small-scale features we provide a new Fig. 5 to illustrate differences between OSTIA data and fine-resolution observations. The problems with accounting for the fine resolution in the coastal zone are used to bridge the results in this section with the ones where we discuss simulations of near-coastal zone and estuaries.

Section 4.3 – it would be helpful to better link this section in to the preceding discussion. The key question, is how does the estuary-specific configuration interface with the larger-scale German Bight models, if at all, and what are the challenges to address in nesting right across scales from North Sea to estuary scale? Is this a 'solved' issue? It is currently difficult to understand how the Ems Estaury model fits relative to other tools available to provide services.

Authors:

- The title of this section (now section 2.5) has been changed.
- The first part of this section was written new, addressing the comment of the referee.
- The focus of the presentation of results has been also changed accordingly, to link this section in to the preceding discussion about the consistence between the estuary-specific modelling and the larger-scale German Bight model.
- More weight in the revised section is given to the transformation of fresh water in the estuary and beyond.

Section 4.6 – please provide some context for the quantitative differences discussed. E.g. is a 40cm difference important for end-users and responding to natural hazards? How do underestimation of 30cm relative to gauge compare with long-term statistics for sea level predictions in this region – is this specific to extreme events or typical?

Authors: In the revised manuscript we mention (end of section 2.4) that "The uncertainties in storm surge predictions and the quantification of associated coastal hazards is of great interest for both short-term forecasts and climate change analyses. Although storm surge forecasting technology is gradually improving, the real-time assessment of the storm surge and inundation area fails to satisfy various demands, particularly for real-time storm forecasting. To reduce the uncertainty of forecasts, knowledge about the processes, such as tide-wave-surge interactions, must be improved. Improved weather forecasting and more adequate coupling between the atmosphere, ocean and waves should further reduce the uncertainty. The use fine horizontal resolution in near-coastal areas, which recently became possible because of the availability of improved computational resources, has proved beneficial. The results of our experiments showed

that the wave-dependent approach, which is not routinely used operationally, yields an ~30% larger surge during the period of "Xavier"."

P2, Para including line 20: ". . .similar devastations never happened again.". Please consider addressing the language in this sentence to something like . . .." similar devastation has not occurred since" – there are of course a number of reasons for this (e.g. have similar magnitude storms hit the region since?).

Authors: We rephrased this sentence.

*P3*, line 25 – please check language concerning "data problematics", suggest rephrasing this point.

Authors: We rephrased this sentence.

P9, line 5 – please clarify status of "in preparation" paper ahead of publication

Authors: We removed this reference.

**Answers of reviewers' comments on "Ocean Forecasting: From Regional to Coastal Scales" by Emil V. Stanev et al.**

**Anonymous Referee #2**

We are grateful to reviewer for the appreciation of some parts of our work and his constructive comments (in italic below), which we answer point-by-point. Some comments of technical character, which are not included in our answer, are addressed in the revised manuscript as the reviewer suggested.

My main objection to the present ms is that it lacks structure and does not present anything really new. Cursory examples are presented, with little to no supporting longterm statistics to back up the various conclusions. As a review paper it is too focused on the shallow water dynamics in one particular region and its very general title is not justified.

Authors:

- In the revised manuscript we improve the structure of the paper with developing the logical links between its sections, as this was proposed also by the first referee.
- We admit that an impression (missing novelty) could have occurred because we did not enough stress on what the new development is. In the revised paper we made clear what the novelties are.
- In the revised manuscript we provide new statistics in form of tables and graphics to support our conclusions.
- We made clear in the revised manuscript that the paper is about short-term predictions, not long-term ones.
- This paper has been submitted to the COSYNA special issue of Ocean Science. Therefore we focus on the areas where most of activities of COSYNA take place that is in "*one particular region*". We want also to mention that it is not possible in one paper to address in sufficient detail many different coastal areas. Our choice was one area, but several different aspects.
- Following the comments of referees we changed the title.

**I recommend ... that the authors instead resubmit a more focused study on the shallow water dynamics in this region, with more emphasis on verification and less emphasis on specific examples.**

Authors:

- The first submission was exclusively on the shallow water dynamics.
- We provide in the revised manuscript more verification material such as Tables 2, 3, 4 providing numbers for statistics, and respective discussion of these new results.
- The number of examples considered has been reduced.

- The presentation is confusing and the text is not properly structured. Again, restricting focus to one specific dynamical problem would help increase the clarity of the presentation. Incosistent use of abbreviations adds to the confusion (e.g. "SAR" vs "search

and rescue").

Authors:

- In the revised manuscript we develop the logical links between individual parts of the paper.
- We formulate for each sub-section more clearly what is the specific dynamical problem addressed.
- The presentation has been restructured (sub-sections omitted, other subsections displaced, some sub-sections are restructured, in some others more weight has been given to issues suggested by the three referees).
- We avoid using misleading abbreviations.

- Central information about the various modeling systems is only given in the appendix. The level of detail is unsatisfactory and the ms cannot stand on its own in its present form. A proper model comparison will require a more elaborate discussion about their differences, for instance the impact of using hourly vs six hourly atmospheric forcing.

Authors:

- As suggested also by the referee #1, we present in the revised manuscript a table of models used in the paper and their most important details.
- We increase (wherever necessary) the level of details given for the individual models in order to balance the deepness of presentation of all models almost the same.
- In the revised manuscript (secton 2.2.2 The rationale of inter-comparison: Numerical models used) we explain the intercomparison strategy. We want to remind that outputs of some operational models have been used and it is not possible to change the way how atmospheric forcing has been applied.
- Our strategy was to use forcing data with as fine as possible resolution in time (see the summary table).

**It is also difficult to keep track of which model is used for what purpose as the authors jump back and forth between them in the examples.**

Authors: We checked carefully all sections and wherever this has not been explained clearly, we provide the necessary information.

**- Errors of representativeness, which becomes an important issue when downscaling data assimilative models merits a discussion, but is not mentioned here.**

Authors: In the revised manuscript we provide new skill estimates (in particular as far as data assimilation is concerned ) and refer to previous studies on this.

- References are missing in several places, e.g. pages/lines 2/15, 3/24, 7/12; there are several errors in the citations (e.g. 5/24, 10/28); and reference to unpublished material makes no sense (9/4).

Authors: We are thankful for this comment and we did all proposed corrections in the revised manuscript.

- The HF radar assimilation technique based on the method of Stanev et al (2015) may be well justified for use in this region, but might be less useful in regions where baroclinicity and/or the influence of complex topography dominates.

**Authors:**

We focus on the German Bight in this paper. Addressing other regions would increase the diversity of addressed issues, which we, following referees' comments want to avoid. Following this comment we mention in the text that "The above demonstration of skill is valid for the specific area. Further application of the proposed method to different regions (e.g., regions dominated by pronounced baroclinicity) requires additional analysis." As said above, in the limited space of this paper, we cannot address many different areas, also because availability of HF radar data is a problem.

**It would be good**

to see an assessment of the impact of HF radar DA on storm surge predictions instead of the (very short) discussion about search-and-rescue support.

Authors: In the revised paper we address the capability of HF radar data to detect changes in surface currents during storm surge periods. In this way we also increase the inter-connectivity between different sections.

**Several published**

papers deal with the impact of HF radar data assimilation on current predictions, e.g. Barth et al (2008, JGR, using ADCP for verification), Yaremchuck et al (2016, DSR II, using drifters), Sperrevik et al (2015, OS, using both drifters and ADCP), so that the cursory example presented here does not really provide anything new.

**Authors:**

- We stressed in the first submission that the novelty is in the forecasting at intra-tidal time scales. Most of the past studies use de-tided HFR data. In the revised manuscript we emphasize on this novelty (as this was one of the suggestions in the general comments).
- In section 3.2 we include a short presentation of earlier works in this field and cite the proposed ones.
- We do not share the statement "*anything new*" and hope that this impression was due to presentation problems with the first submission. Furthermore, what has been described here is in the hearth of one (among not too many) continuously operational system assimilating HF radar data, which is put in place in the frame of COSYNA (the theme of this special issue).

- The apparently small impact of in-situ data (ferrybox) vs the OSTIA product indicates that the DA system is not working optimally. I would expect in-situ data to be rather

more valuable, but again, very little in the way of statistics is presented, e.g. innovations vs analysis increments and their temporal and spatial distributions. Mention could also be made about rapid update cycles, which is used successfully by e.g. the KNMI in their regional NWP system to maximise the use of observations in small model domains (deHaan, 2013, QJRMS).

Authors:

- We explain in the revised manuscript the reasons of "*small impact of in-situ data*" providing also more statistics.
- We show a comparison between OSTIA and fine-resolution temperature data.
- We present in the revised manuscript more statistics and refer to skill estimates analyzed in similar earlier publications (Grayek et al., 2011; Stanev et al., 2011) where more details are given about the systems' performance.
- We refer in the revised manuscript to the publication mentioned by the referee.

- The "two-way nesting" method described in Sec. 2.3 differs from the full online nesting implemented in e.g. ROMS and AGRIF. I assume the nudging based method presented here will in practice work as a low pass filter when information is exchanged between parent and child grids, and I would like to see what the impact is on fast time scales such as tidal wave propagation.

Authors: Answering this question, we provide in the revised manuscript more details about temporal variability (tidal analysis and variability of salinity in the transition area), which demonstrates that the proposed method does not impact negatively the dominant dynamics in the transition area.

**Answers on reviewers' comments on "Ocean Forecasting: From Regional to Coastal Scales" by Emil V. Stanev et al.**

**Anonymous Referee #3**

We are grateful to reviewer for the appreciation of some parts of our work and his constructive comments (in italic below), which we answer point-by-point.

**The manuscript contains a lot of information (from Data assimilation to tides, wave – current interactions estuarine and search & rescue applications) which in most situations is not well structured/organized and sometimes becomes quite confusing for the reader.**

Authors: In the revised manuscript substantial restructuring has been done. We explain better the links between individual parts of paper. We removed sub-sections 4.1 and 4.4, moved sub-section 4.5 into the new section 3. Old section 4.6 was moved into the modelling part (new section 2). We also changed the titles of some sub-sections.

**Moreover no mention at all for the effect of**

atmospheric forcing in coastal forecasting is given. I think a whole subsection should be devoted to this important for coastal applications aspect. Along this line air-sea interaction and issues related to wave current interactions (for example the momentum and energy surface boundary condition) should be discussed in more detail.

**Authors:**

There are some reasons not to devote one separate section to the atmospheric forcing. In the revised manuscript, we add some references to studies on this subject carried out in the past (Backhaus, 1989; Skogen et al., 2011, Dangendor et al., 2014). Another argument for this follows the suggestions to keep this manuscript more focused. The third one was to demonstrate novel developments, and we consider the issue about shallow-water tides as one such issue. In the revised paper we integrated the issue of atmospheric forcing with the novel development of coupled wave-current modelling. Additionally, more detailed presentation on the coupling method is given.

**I think**

that the authors should concentrate on mostly 2-3 topics (for example data assimilation of HF Radar or satellite/in-situ SST data on the coastal scale and wave –current interactions) instead of overwhelming the reader with excessive material which is not complete (for example in section 4.6 where the important topic of wave – current interactions is involved/discussed the reader is just referred the paper by Staneva et al., 2015 for the scientific approach & discussion) and cannot be easily digested.

**Authors:**

- In the revised manuscript we provide more coherent and complete presentation of the material.
- Section 4.6 has been reshaped in lines with what referee proposes.

In this sense, I propose a major revision of the present manuscript with drastic restructuring and focusing on a much more limited list of topics related to coastal forecasting.

Authors: As said above, we restructured the paper, removed parts (which are not so closely related to the coastal forecasting) and re-focused, as suggested by referee. To our understanding all aspects considered in the resubmitted paper are linked to coastal forecasting, and we hope that we expressed this in a more convincing way now.

-The title of the manuscript should contain the toponym "German Bight". I agree with the new title proposed by the anonymous referee #1

Authors: The title has been changed as suggested by referee#1: "Ocean Forecasting for the German Bight: From Regional to Coastal Scales".

-Section 3.1: The approach proposed to overcome the situation where the assimilation degrades the model results due to hf perturbations, is never presented explicitly in this paper.

Authors: We present this issue in the revised manuscript.

-Section 3.3: what do we see in fig. 5b? The analysis RMSE? If yes I would prefer to judge the performance of the assimilation system by checking the forecast RMSE.

Authors: New statistical analysis is provided here to demonstrate the skill of data assimilation.

In any case a more in depth analysis of the results is needed in order to understand the impact of OSTIA and in-situ observations.

Authors: Thank you for this suggestion. We address this issue in more detail in the revised manuscript.

-Section 4.4 can be omitted. I do not understand its role in this manuscript.

Authors: We omitted this section.

-Section 4.6: more in depth presentation and analysis of the results is needed.

Authors: We substantially revised this part presenting in more detail the coupled model and the analysis of simulations.

---

## Author Response (AR2)

Dear Editor and Springer,

Thank you for accepting this manuscript for publication.

Emil Stanev